# Yeast derlin Dfm1 employs a chaperone-like function to resolve misfolded membrane protein stress

Rachel Kandel[1], Jasmine Jung[1], Della Syau[1], Tiffany Kuo[1], Livia Songster[1], Casey Horn[1], Claire Chapman[1], Analine Aguayo[1], Sascha Duttke[2], Christopher Benner[3], Sonya E. Neal [1] *

1 Division of Biological Sciences, the Section of Cell and Developmental Biology, University of California San Diego, La Jolla, California, United States of America, 2 School of Molecular Biosciences, College of Veterinary Medicine, Washington State University, Pullman, Washington, United States of America, 3 Department of Cellular and Molecular Medicine, University of California San Diego, La Jolla, California, United States of America

* seneal@ucsd.edu

**Data Availability Statement:** Code for microscopy puncta analysis is available at https://github.com/LiviaSongster/yeast-fluor-percent-puncta. Code for

## Abstract

Protein aggregates are a common feature of diseased and aged cells. Membrane proteins comprise a quarter of the proteome, and yet, it is not well understood how aggregation of membrane proteins is regulated and what effects these aggregates can have on cellular health. We have determined in yeast that the derlin Dfm1 has a chaperone-like activity that influences misfolded membrane protein aggregation. We establish that this function of Dfm1 does not require recruitment of the ATPase Cdc48 and it is distinct from Dfm1's previously identified function in dislocating misfolded membrane proteins from the endoplasmic reticulum (ER) to the cytosol for degradation. Additionally, we assess the cellular impacts of misfolded membrane proteins in the absence of Dfm1 and determine that misfolded membrane proteins are toxic to cells in the absence of Dfm1 and cause disruptions to proteasomal and ubiquitin homeostasis.

## Introduction

While misfolded proteins are recognized as a source of cellular stress, the mechanisms by which cells prevent this stress and how this stress impacts cells is not fully understood. Eukaryotic cells are equipped with protein quality control pathways for preventing the accumulation of aggregation-prone misfolded proteins. The endoplasmic reticulum (ER) is responsible for folding both secretory and membrane proteins and is well equipped with quality control pathways for refolding or eliminating misfolded proteins. One of the major pathways of protein quality control at the ER is ER-associated degradation (ERAD) [1]. ERAD utilizes the ubiquitin proteasome system (UPS) to selectively target and degrade misfolded or unassembled proteins at the ER [2].

ERAD is a well-conserved process from yeast to mammals. ERAD of membrane proteins requires 4 universal steps: (1) substrate recognition [3–6]; (2) substrate ubiquitination [7]; (3)

RNA sequencing analysis is available at Mendeley Data, doi: 10.17632/scr83vh7vb.2.

**Funding:** This work was supported by the National Institutes of Health (1R35GM133565 to SEN; R00GM135515 to SD), the National Science Foundation (2047391 to SEN), and the Pew Charitable Trusts (34089 to SEN). The funders had no role in study design, data collection and analysis, decision to publish, or preparation of the manuscript.

**Competing interests:** The authors have declared no competing interests exist.

**Abbreviations:** AATD, alpha-1 antitrypsin deficiency; A1PiZ, alpha-1 proteinase inhibitor; CFTR, cystic fibrosis transmembrane receptor; CFU, colony-forming unit; DDM, dodecyl maltoside; DOA, degradation of alpha2; ER, endoplasmic reticulum; ERAD, endoplasmic reticulum-associated degradation; EV, empty vector; GO, gene ontology; HRP, horseradish peroxidase; IPB, immunoprecipitation buffer; PACE, proteasome-associated control element; PCA, principal component analysis; PCR, polymerase chain reaction; PI, protease inhibitor; RNA-seq, RNA sequencing; SUS, self ubiquitinating substrate; TCA, trichloroacetic acid; TMD, transmembrane domain; UPR, unfolded protein response; UPS, ubiquitin proteasome system.

retrotranslocation of substrate from the ER to the cytosol [8–13]; and (4) degradation by the cytosolic proteasome [2,14,15]. A hexameric cytosolic ATPase, Cdc48 in yeast and p97 in mammals, is required for retrotranslocation of all ERAD substrates [16–18]. In the context of this paper, substrate refers to a protein that is targeted by the ERAD pathway.

In yeast, ER membrane substrates can be targeted by the DOA (degradation of alpha2) pathway or the HRD pathway (hydroxymethyl glutaryl-coenzyme A reductase degradation), utilizing the E3 ligases Doa10 and Hrd1, respectively. Additionally, the yeast derlin Dfm1 is specifically required for the retrotranslocation of misfolded membrane substrates, in both the HRD and DOA pathways [8]. Dfm1 facilitates retrotranslocation of membrane proteins through several mechanisms including (1) recognition and binding to misfolded membrane proteins; (2) thinning the lipid bilayer to reduce the thermodynamic barrier to extraction; and (3) recruiting the ATPase Cdc48 to the ER [8,19].

Dfm1 is a member of the derlin subclass of rhomboid proteins. Rhomboid proteins are a widely conserved family of proteins, found in all domains of life [20–23]. There are 2 major categories of rhomboid proteins: active rhomboid proteases and inactive rhomboid pseudo-proteases. While the inactive rhomboid pseudoproteases lack a catalytic site, they have been implicated in a wide variety of biological processes, including protein quality control, protein trafficking, and cell signaling [24–28]. Derlin proteins, including Dfm1, are rhomboid pseudo-proteases that are critical for ERAD of a wide variety of substrates, both in yeast and mammalian cells [8,29–32].

We have previously observed that in *dfm1Δ* cells, when a misfolded membrane protein is strongly expressed, the cells show a severe growth defect [33]. This is seen specifically in the absence of Dfm1, and this growth defect is not observed in the absence of other ERAD components, indicating a specific function for Dfm1 in sensing and/or adapting cells to misfolded membrane protein stress (Fig 1) [33]. This is in line with a previous study linking Dfm1 to ER homeostasis [34].

In the present study, we determine that Dfm1 prevents membrane protein toxicity because of a previously unidentified chaperone-like function that is independent of Cdc48 recruitment. This function is distinct from Dfm1's role in protein retrotranslocation, while also relying on many of the same functions deployed by Dfm1 to promote retrotranslocation. We further determined that human homologs of Dfm1 have also retained this ability. This study is the first to demonstrate chaperone-like activity for any rhomboid protein. Many rhomboid proteins use similar functions to Dfm1 to promote retrotranslocation, and the rhomboid protease RHBDL4 has recently been characterized as acting on aggregation-prone substrates [19,28,35–37]. It will be an interesting and important further line of inquiry to determine if a chaperone-like activity is a common ability of rhomboid proteins, both for the pseudoproteases and proteases.

As a complement to our work on the function of Dfm1 in relieving misfolded membrane protein toxicity, we also sought to determine how misfolded membrane proteins cause toxicity. We determine that misfolded membrane proteins, but not other types of misfolded proteins, impact proteasome and ubiquitin homeostasis. We also identified several proteins that promote cellular health upon misfolded membrane protein by resolving the proteasome and ubiquitin stress that misfolded membrane proteins trigger. Intriguingly, we also find that not all membrane protein aggregates are toxic. The combination required for toxicity appears to be both (1) aggregated misfolded membrane proteins; and (2) ubiquitinated misfolded membrane proteins. Either of these features alone is not sufficient for toxicity.

We propose a model in which upon accumulation of ubiquitinated misfolded membrane proteins in the absence of Dfm1, misfolded membrane proteins form toxic aggregates. In the presence of Dfm1, this toxicity is prevented by Dfm1's ability to solubilize membrane proteins, independent of its ability to retrotranslocate proteins.

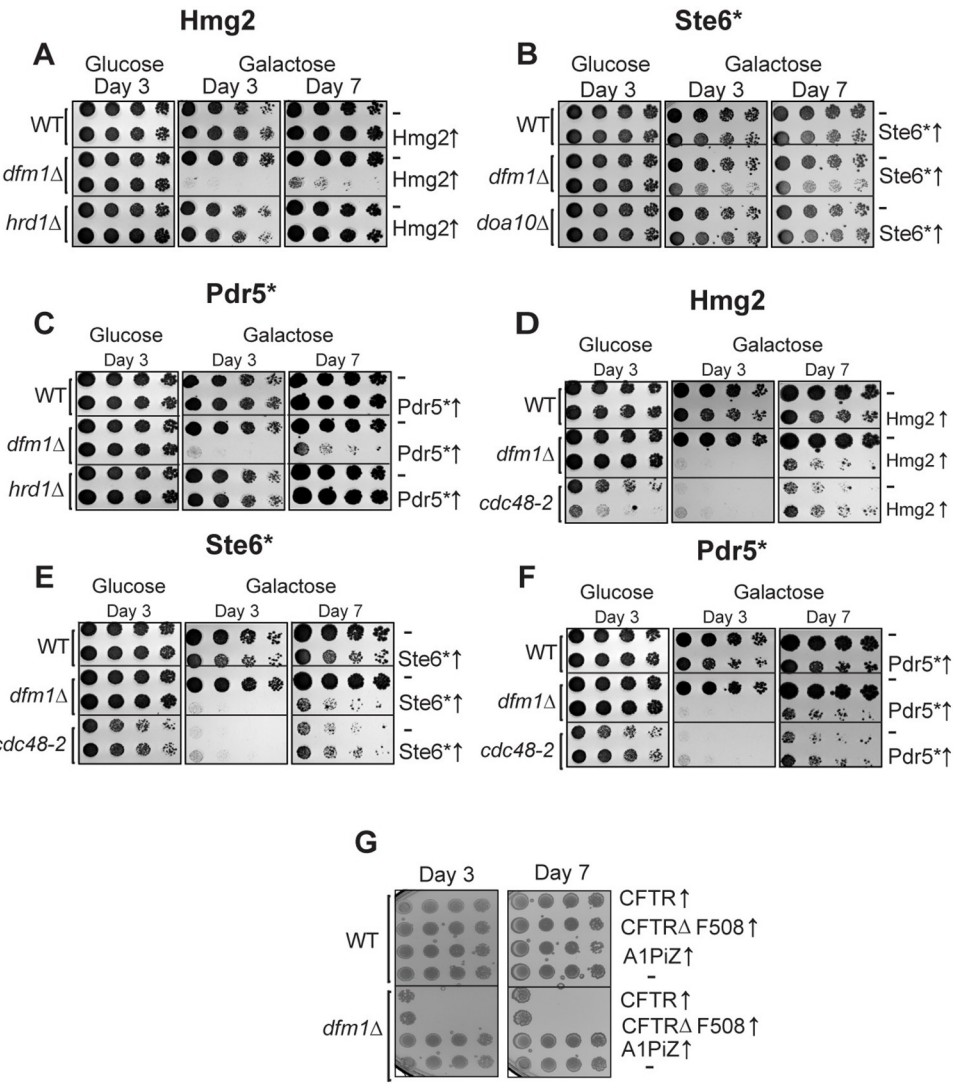

**Fig 1. Integral membrane protein overexpression causes a growth defect in *dfm1Δ* cells in an ERAD independent manner. (A)** WT, *dfm1Δ*, and *hrd1Δ* cells containing either GAL_pr-HMG2-GFP or EV were compared for growth by dilution assay. Each strain was spotted 5-fold dilutions on glucose or galactose-containing plates to drive HMG2-GFP overexpression, and plates were incubated at 30˚C. **(B)** Dilution assay as described in (A) except using WT, *dfm1Δ*, and *doa10Δ* cells containing either GAL_pr-STE6*-GFP or EV. **(C)** Dilution assay as described in (A) except using WT, *dfm1Δ*, and *hrd1Δ* cells containing either GAL_pr-PDR5*-HA or EV. **(D)** Dilution assay as described in (A) except using WT, *dfm1Δ*, and *cdc48-2* cells. **(E)** Dilution assay as described in (B) except using WT, *dfm1Δ*, and *cdc48-2* cells. **(F)** Dilution assay as described in (C) except using WT, *dfm1Δ*, and *cdc48-2* cells. **(G)** Dilution assay as described in (A) except using WT or *dfm1Δ* cells expressing human CFTR, CFTRΔF508, or A1PiZ and plated only on glucose-containing plates. All dilution growth assays were performed in 3 biological and 2 technical replicates (*N* = 3). CFTR, cystic fibrosis transmembrane receptor; ERAD, endoplasmic reticulum-associated degradation; EV, empty vector.

## Results

### Absence of Dfm1 and expression of integral misfolded membrane proteins causes growth stress

Previous research from the Neal lab has revealed that accumulation of a misfolded membrane protein in the absence of Dfm1 causes a severe growth defect in the substrate-toxicity assay [33]. In the substrate-toxicity assay, yeast strains with a misfolded protein under the control of

a galactose inducible promoter are plated in a spot assay onto selection plates with either 2% galactose or 2% dextrose as a carbon source (Fig 1) [38]. This allows for comparison of growth of yeast strains with different genetic perturbations with expression of misfolded substrates. This growth defect can be seen with strong expression of 3 misfolded membrane proteins in *dfm1Δ* cells: Hmg2, Pdr5*, and Ste6* (Fig 1A–1C). We have previously shown that this growth defect is specific to misfolded membrane proteins at the ER, as expression of a luminal ERAD substrate, CPY*, in *dfm1Δ* cells elicits no growth defect [33]. Interestingly, this growth defect is not seen when membrane proteins accumulate in the absence of other ERAD components, such as the E3 ligases Hrd1 and Doa10 (Fig 1A–1C). In the case of *dfm1Δ*, *hrd1Δ*, and *doa10Δ* cells, misfolded membrane proteins accumulate at the ER due to defects in ERAD, but only in the case of *dfm1Δ* cells, is a growth defect observed with misfolded membrane protein expression. Altogether, we surmise that this growth defect triggered by the absence of Dfm1 along with expression of misfolded membrane protein is due to cellular stress caused by misfolded membrane protein toxicity.

By utilizing the substrate-toxicity assay, we observed a growth defect in *dfm1Δ* cells and normal growth in *hrd1Δ* and *doa10Δ* cells upon expression of ERAD membrane substrates (Fig 1A–1C). The cell biological difference among these ERAD knockout strains is that membrane substrates are ubiquitinated in *dfm1Δ* cells but not ubiquitinated in *hrd1Δ* and *doa10Δ* cells, due to the absence of the ER E3 ligases, as determined through western blot for ubiquitin (Fig 7B). One possibility is that the growth stress is not specific to *dfm1Δ* cells and is solely dependent on the accumulation of ubiquitinated membrane substrates. To rule out this possibility, we utilized a temperature-sensitive Cdc48 allele strain, *cdc48-2*, which, like *dfm1Δ* cells, results in accumulation of ubiquitinated ERAD membrane substrates [8]. While we used *cdc48-2* cells at the permissive temperature of 30°C, ERAD is still compromised for Hmg2, as previously reported, and we validated using a cycloheximide chase that Pdr5* and Ste6* degradation is also impaired at 30°C in *cdc48-2* cells (S1A and S1B Fig) [8,39]. The substrate-toxicity assay was employed on *cdc48-2* strains expressing membrane substrates Hmg2, Pdr5*, and Ste6* (Fig 1D–1F). These strains showed a growth defect while growing on galactose plates due to inherent slow growth of *cdc48-2* strains, but this was not worsened by expression of misfolded integral membrane proteins, despite the E3 ligases that ubiquitinate these proteins still being present. These results indicate that Dfm1 plays a specific role in the alleviation of misfolded membrane protein stress.

## Disease-associated membrane proteins cause growth stress

Since a wide variety of misfolded membrane proteins elicit growth stress in *dfm1Δ* cells, we hypothesized that growth stress would also be observed with expression of clinically relevant human misfolded membrane proteins. We tested expression of WT cystic fibrosis transmembrane receptor (CFTR), CFTRΔF508, the most common disease-causing variant of CFTR, and the Z variant of alpha-1 proteinase inhibitor (A1PiZ), a protein variant that results in alpha-1 antitrypsin deficiency (AATD). All these substrates are targeted by ERAD when expressed in yeast, with CFTR and CFTRΔF508 being targeted by machinery for membrane proteins and A1PiZ being a soluble misfolded protein in the ER lumen [40,41]. When these proteins were expressed in *dfm1Δ* cells, both CFTR and CFTRΔF508 resulted in a growth defect, while none was observed with expression on A1PiZ (Fig 1G). Expression of any of the proteins in WT yeast cells resulted in no growth defect (Fig 1G). It was not wholly surprising that WT CFTR also elicited growth stress in *dfm1Δ* cells. Previous studies have shown that while virtually all CFTRΔF508 is targeted to ERAD, about 80% of WT CFTR is degraded via ERAD in yeast and mammals [40,42,43].

## Dfm1 has a dual role in ER protein stress and ERAD retrotranslocation

Previous work from the Hampton lab establishing a role for Dfm1 in misfolded membrane protein retrotranslocation also identified several motifs of Dfm1 that are essential for its retrotranslocation function [8]. Additionally, by employing an unbiased genetic screen, our lab recently identified 5 residues of Dfm1 that are required for retrotranslocation [19]. Here, we tested whether these residues, critical for Dfm1's retrotranslocation function, are required for alleviating the growth stress in *dfm1Δ* cells expressing Hmg2.

Fig 2A shows a schematic of Dfm1, with the regions of the protein important for retrotranslocation function highlighted, and a list of specific motifs and residues that are mutated listed in a table [8,19]. Dfm1 contains 2 motifs that are well conserved among the rhomboid superfamily, the WR motif in Loop 1 and the GxxxG (Gx3G) motif in transmembrane domain (TMD) 6 [8,27]. Both of these motifs are required for Dfm1-mediated retrotranslocation [8,44]. We first tested the requirement of the conserved rhomboid motif mutants by expressing Hmg2 with WR mutants (WA and AR) and Gx3G mutants (Ax3G and Gx3A) and observed no restoration in growth (Fig 2B). Our previous work determined that Loop 1 mutants (F58S, L64V, and K67E) obliterated Dfm1's ability to bind misfolded membrane substrates, and TMD 2 mutants (Q101R and F107S) reduce the lipid thinning ability of Dfm1, a function that aids in Dfm1's retrotranslocation function [44]. Accordingly, we utilized these mutants in our growth assay and did not observe a rescue of the growth defect (Fig 2C). We have previously shown that alteration of the 5 signature residues of the Dfm1 SHP box to alanine (Dfm1-5Ashp) ablates its ability to recruit Cdc48 (Fig 2D). We also established, Dfm1's Cdc48 recruitment function is required for Dfm1's retrotranslocation function, whereas the Dfm1-5Ashp mutant impairs its retrotranslocation function [8]. Notably, in contrast to the other mutants tested, Dfm1-5Ashp was still able to alleviate the growth defect like WT Dfm1 (Fig 2E). These results suggest that Dfm1's substrate engagement and lipid thinning function is required for alleviating membrane substrate-induced stress, whereas Dfm1's Cdc48 recruitment function is dispensable for alleviating the growth stress. We validated that expression of WT Dfm1 and all Dfm1 mutants were comparable using western blot (S1C Fig).

## Human derlins relieve growth stress

Dfm1 is a rhomboid pseudoprotease and a member of the derlin subclass of rhomboid proteins. The human genome encodes 3 derlins, Derlin-1, Derlin-2, and Derlin-3. Yeast Dfm1 is the closest homolog of the mammalian derlins [27]. All 3 are ER-localized proteins that are implicated in ERAD and adaptation to ER stress [29,30,45–48]. We expressed human Derlin-1 and Derlin-2 in *dfm1Δ*+Hmg2 cells. We opted to only test Derlin-1 and Derlin-2, as they are more structurally similar to Dfm1 and also contain cytoplasmic SHP box motifs [19]. Both human derlins were able to rescue growth in these cells in the substrate-toxicity assay (Fig 2F). This was surprising, as we had previously found that mammalian derlins cannot complement the retrotranslocation function of Dfm1 in yeast cells for self-ubiquitinating substrate (SUS)-GFP, a similar ERAD substrate to Hmg2 [19].

## Dfm1 solubilizes misfolded membrane protein aggregates independent of Cdc48 recruitment

The above studies show that Dfm1 residues critical for retrotranslocation—through substrate binding and its lipid thinning function—are also important for alleviating membrane substrate-induced stress. Conversely, add-back of a Dfm1 Shp box mutant (Dfm1-5Ashp) (Fig 2D), which does not recruit Cdc48 and cannot retrotranslocate proteins, is able to restore

growth in the substrate-toxicity assay. We surmise that Dfm1's actions—independent of its Cdc48 recruitment function—may be directly acting on misfolded membrane substrates to prevent growth stress. One possibility is that Dfm1 may directly act on misfolded membrane substrates by functioning as a chaperone-like protein to prevent misfolded membrane protein toxicity. We hypothesize that Dfm1 acts as either a holdase, preventing the aggregation of misfolded membrane substrates, or as a disaggregase, separating proteins in existing protein aggregates. To address this hypothesis, we employed a detergent solubility assay in *dfm1Δ* +Hmg2 cells with add-back of WT DFM1 or DFM1 mutants. ER microsomes were isolated and incubated in 1% dodecyl maltoside (DDM) and subjected to centrifugation to separate aggregated substrate (pellet fraction) from non-aggregated substrate (soluble fraction). As shown in Fig 3A, nearly all Hmg2-GFP in *dfm1Δ* cells was pelleted (aggregated). Conversely, with Dfm1 and Dfm1-5Ashp add-back cells, nearly all Hmg2-GFP was solubilized (non-aggregated). This striking all-or-nothing phenotype of Hmg2 aggregation demonstrates an important role for Dfm1 in influencing membrane protein aggregation. As a control for these studies, we examined Hmg2 in both WT, *hrd1Δ* cells, and *hrd1Δ*+Hrd1 cells. Nearly all protein was soluble in all 3 strains (Fig 4A). We also tested a properly folded ER membrane protein, Sec61-GFP in *dfm1Δ* cells. In contrast to Hmg2-GFP, majority of Sec61-GFP was in the detergent-solubilized supernatant fraction and there was no change in Sec61-GFP detergent solubility with Dfm1 or Dfm1-5Ashp add-back in *dfm1Δ* cells (Fig 4B).

It appears Dfm1—independent of its Cdc48 recruitment function—functions as a chaperone-like protein to influence the aggregation of misfolded membrane proteins. We next explored additional Dfm1 residues that are required for solubilizing membrane substrates. Accordingly, mutants in the conserved rhomboid motifs (AR and Ax3G) were employed in the detergent solubility assay. DFM1-AR and DFM1-Ax3G add-back resulted in aggregated HMG2 (Fig 3A). Similarly, retrotranslocation defective Dfm1 mutants in Loop 1 (F58S, L64V, and K67E) and TMD 2 mutants (Q101R and F107S) in the detergent solubility assay were not capable of solubilizing Hmg2 (Fig 3B). This all-or-nothing effect that Dfm1's presence has on aggregation led us to determine whether Dfm1 binds to Hmg2 even after solubilization with DDM. Indeed, using co-immunoprecipitation, we found that Dfm1 physically interacts with solubilized Hmg2 (Fig 3C). Altogether, Dfm1 is critical in influencing the aggregation state of its ERAD membrane substrate (Fig 3D). Although the ability to recruit Cdc48 is vital for Dfm1's retrotranslocation function, it is not required for this newly established chaperone-like function.

The chaperone-like function of Dfm1 is generalizable to other misfolded membrane proteins but not non-membrane misfolded proteins. We tested other membrane ERAD membrane substrates targeted by the HRD (Pdr5*) or DOA (Ste6*) pathways, and they were both completely solubilized in the presence of Dfm1 in the detergent solubility assay (Fig 4C and 4D). In contrast, the misfolded ER luminal protein, CPY*, was completely solubilized regardless of the presence or absence of Dfm1 (Fig 4E). Notably, we also observed that Derlin-1 and Derlin-2 were able to prevent aggregation of Hmg2 in *dfm1Δ* cells, indicating that other derlin proteins have a conserved chaperone-like function (Fig 4F).

We also investigated whether Hmg2-GFP appeared in puncta in *dfm1Δ* cells through confocal microscopy. We found no significant difference between the percentage of GFP in puncta or the number of puncta with add-back of WT DFM1 or any of the DFM1 mutants in *dfm1Δ* cells (S1D–S1F Fig). This is in line with the view of some in the field that toxic aggregates are generally below the visible detection limit for confocal microscopy and that puncta identified through microscopy tend to be representative of sequestrosomes, a cellular adaptation to the accumulation of aggregation prone proteins [49].

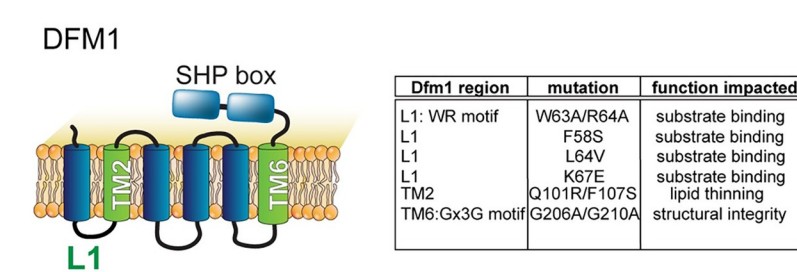

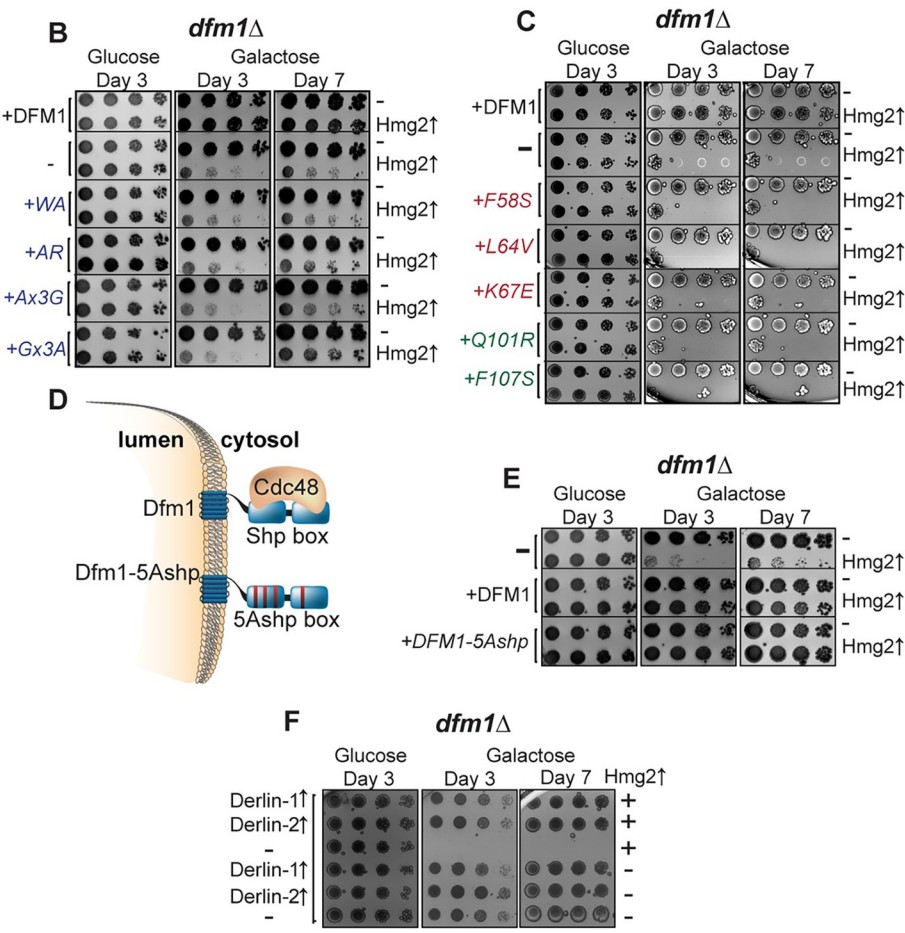

**Fig 2. Dfm1 retrotranslocation defective mutants show differing abilities to restore growth. (A)** Depiction of Dfm1, which highlights L1, TM2, TM6, and its SHP box domain. The table indicates the Dfm1 region, amino acid mutation, and the corresponding function that is specifically impaired. All mutants have been previously identified as being required for retrotranslocation and when mutated did not restore growth in *dfm1*Δ cells expressing an integral membrane protein (GAL$_{pr}$-HMG2-GFP). **(B)** *dfm1*Δ cells with an add-back of either WT DFM1-HA, EV, DFM1-WA-HA, DFM1-AR-HA, DFM1-Ax3G-HA, or DFM1-Gx3A-HA containing either GAL$_{pr}$-HMG2-GFP or EV were compared for growth by dilution assay. Each strain was spotted 5-fold dilutions on glucose or galactose-containing plates to drive Hmg2-GFP overexpression, and plates were incubated at 30°C. **(C)** Dilution assay as described in (B) except using an add-back of either WT Dfm1-HA, EV, Dfm1-F57S-HA, Dfm1-L64V-HA, Dfm1-K67E-HA, Dfm1-Q101R-HA, or Dfm1-F107S-HA. **(D)** Depiction of Dfm1 and Dfm1-5Ashp. Dfm1 is an ER-localized membrane proteins with 6 TMDs. Both versions of Dfm1 have a cytoplasmic shp box, but the 5Ashp mutant is unable to recruit the cytosolic ATPase Cdc48. **(E)** Dilution assay as described in (B) except using add-back of either EV, WT DFM1-HA, or DFM1-5Ashp-HA mutant. **(F)** Dilution assay as described in (B) except with add-back of

human Derlin-1-Myc or Derlin-2-Myc. All dilution growth assays were performed in 3 biological replicates and 2 technical replicates (*N* = 3). ER, endoplasmic reticulum; EV, empty vector; TMD, transmembrane domain.

## Misfolded membrane proteins do not activate the unfolded protein response

The canonical ER stress pathway triggered by the accumulation of misfolded proteins is the unfolded protein response (UPR) [50]. The UPR is known to be induced by the accumulation of misfolded soluble proteins within the ER lumen. To test if misfolded membrane protein accumulation at the ER activates UPR, we used a fluorescence-based flow cytometry assay. In this assay, yeast cells encoding both a galactose inducible misfolded protein or empty vector (EV) and UPR reporter 4xUPRE-GFP were treated with or without 0.2% galactose and 2 µg/mL of the ER stress inducing drug tunicamycin or DMSO as the vehicle control. GFP expression was measured by flow cytometry for 5 hours following galactose treatment. We found that GFP expression did not increase by 5 hours post-galactose addition in *dfm1Δ* cells compared to *pdr5Δ* cells expressing any of the following substrates tested: Hmg2, Ste6*, or EV (Fig 5A–5D, 5G and 5H). *Dfm1Δ* cells are able to activate UPR, as addition of tunicamycin to these cells allowed them to activate the UPR at similar levels as *pdr5Δ* cells (Fig 5A–5H). As expected, expression of the ER luminal substrate CPY* activated the UPR in *dfm1Δ* cells and *pdr5Δ* cells (Fig 5E and 5F).

Our findings from flow cytometry experiments were further corroborated by measuring Hac1 splicing via polymerase chain reaction (PCR) (S2A and S2B Fig). When the UPR is active, the mRNA of the transcription factor Hac1 is spliced to create a transcript 252bp shorter than the full-length transcript [51]. Samples with a band for both spliced and unspliced Hac1 indicated UPR activation, while a single band of the unspliced variant indicated no UPR activation. The results from these experiments were in agreement with the flow cytometry-based assay; we found no HAC1 splicing with misfolded membrane protein overexpression in *dfm1Δ* cells (S2A and S2B Fig).

## Accumulation of misfolded membrane proteins up-regulates proteasome components

After determining the UPR is not activated in *dfm1Δ* cells expressing Hmg2, we next sought to determine the transcriptional changes that occur with misfolded membrane protein stress. To address this question, we utilized RNA sequencing (RNA-seq). We prepared and sequenced cDNA libraries from mRNA extracted from *pdr5Δ* cells, *hrd1Δpdr5Δ* cells, and *dfm1Δ pdr5Δ* cells containing galactose inducible Hmg2 or EV, 2 hours post-galactose treatment. These yeast strains were generated from a yeast knockout collection with the BY4742 strain background, and *pdr5Δ* cells are commonly used as the wild-type background for the knockout collection. We validated using the substrate-toxicity assay that *dfm1Δ pdr5Δ*+Hmg2 strains in this background also display a growth defect (S3A Fig). We used principal component analysis (PCA) to determine genes that were up-regulated and down-regulated most in *dfm1Δ* cells expressing Hmg2 versus the control strains; WT+EV, WT+Hmg2, *hrd1Δ*+EV, *hrd1Δ*+Hmg2, and *dfm1Δ*+EV (S1 Data). Principal component 1 (PC1) value of all replicate strains except for *dfm1Δ*+Hmg2 cells clustered closer to each other than they did to either replicate of the *dfm1Δ*+Hmg2 cells, indicating that these strains were transcriptionally distinct from the others sequenced (S3B Fig). Additionally, the *dfm1Δ*+Hmg2 replicates were fairly distinct from each other, so while there were genes up-regulated in both replicates, there were also variable transcriptional changes (S3B Fig). This variability is likely representative of biological variability in

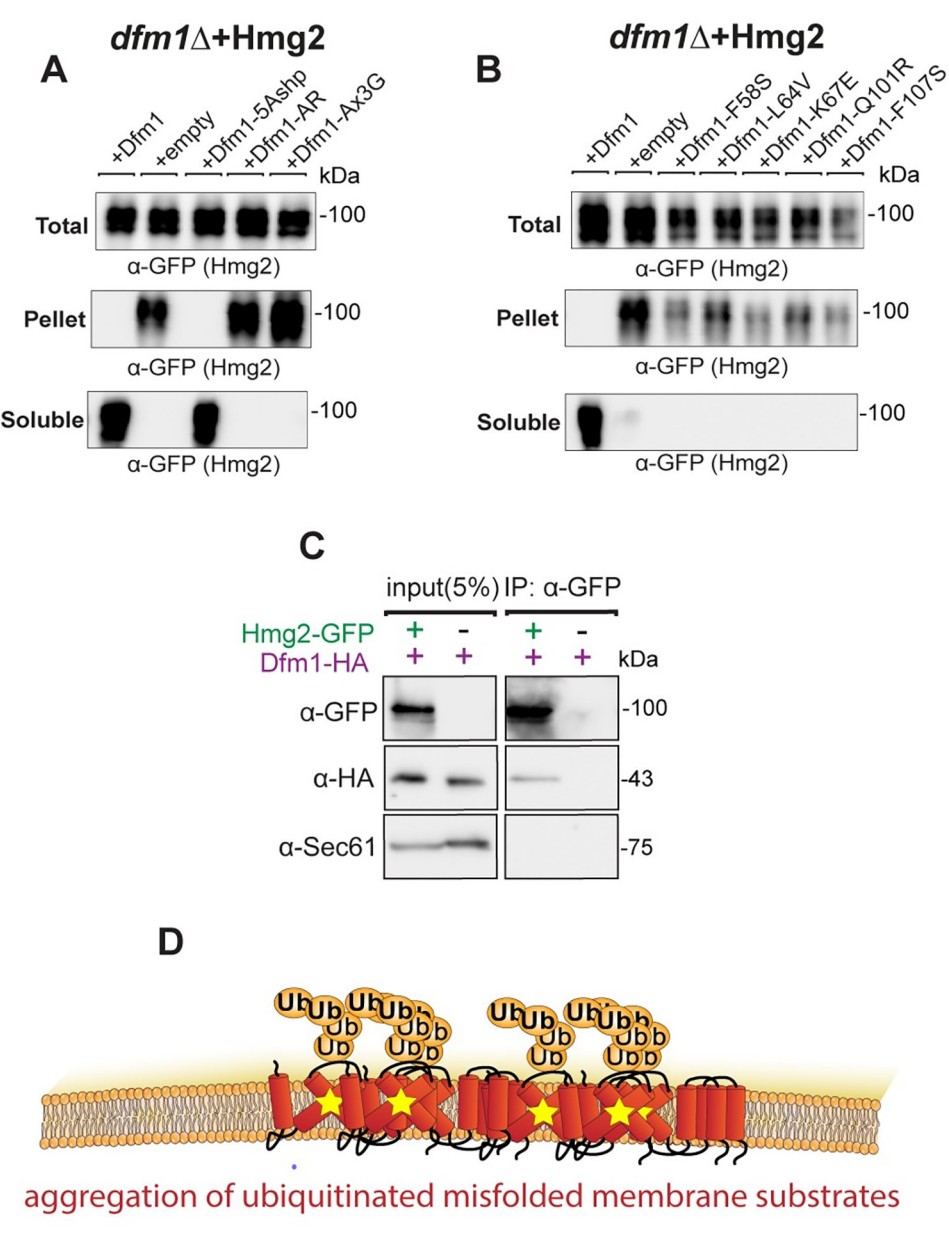

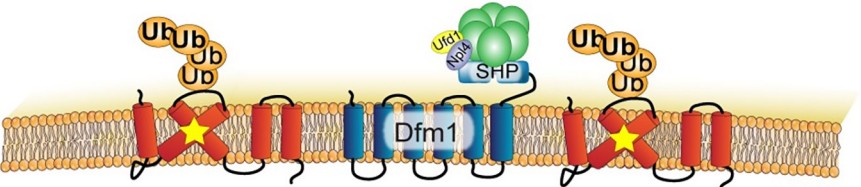

**Fig 3. Dfm1 reduces misfolded membrane protein toxicity through a chaperone-like activity. (A)** Western blot of aggregated (pelleted) versus non-aggregated (soluble) membrane proteins at the ER. Lysates from *dfm1Δ* cells containing HMG2-GFP, with either add-back of WT DFM1-HA, EV, DFM1-5Ashp-HA, DFM1-AR-HA, and DFM1-AxxxG-HA were blotted using anti-GFP to detect Hmg2. Top: Total fraction. Middle: ER pelleted fraction. Bottom: ER DDM solublilized fraction. **(B)** Western blot of aggregated versus non-aggregated membrane proteins at the ER as in (A) but with add-back of either WT DFM1-HA, EV, DFM1-F58S-HA, DFM1-L64V-HA,

DFM1-K67E-HA, DFM1-Q101R-HA, and DFM1-F107S-HA. **(C)** DDM solubilized Hmg2-GFP binding to Dfm1-HA was analyzed by Co-IP, using anti-GFP to detect Hmg2 and anti-HA to detect Dfm1-HA. As negative control, cells not expressing Hmg2-GFP were used. Sec61 was analyzed as another negative control for nonspecific binding using anti-Sec61 (3 biological replicates, $N$ = 3). **(D)** Graphic depicting integrated model of Dfm1's function in misfolded membrane protein stress. Top: Misfolded membrane proteins in the absence of Dfm1 forming aggregates within the ER membrane. Bottom: Cells with WT Dfm1 or 5Ashp-Dfm1 promoting non-aggregated misfolded membrane proteins and preventing cellular toxicity. Data Information: All detergent solubility assays were performed with 3 biological replicates ($N$ = 3). DDM, dodecyl maltoside; ER, endoplasmic reticulum; EV, empty vector.

these strains rather than experimental variability as it was only observed between these replicates and not replicates of the other strains tested.

Up-regulated (+PC1 values) and down-regulated (-PC1 values) genes in *dfm1Δ*+Hmg2 cells were used for gene ontology (GO) analysis. The most overrepresented group of up-regulated genes were those classified as being involved in "Proteasomal Ubiquitin-Independent Protein Catabolic Processes," "Regulation of Endopeptidase Activity," and "Proteasome Regulatory Particle Assembly" (S3D Fig). Several proteasome subunits were represented in this list of up-regulated genes. The most overrepresented group of down-regulated genes in this dataset were those classified as being involved in "rRNA Export from Nucleus," "rRNA Transport," and "Translational Termination" (S3E Fig). Because a down-regulation of the mRNA for genes encoding ribosomal proteins is a general feature of stressed yeast cells [52], we focused on the up-regulation of proteasome components. Plotting the PC1 and PC2 values for *dfm1Δ* +Hmg2 cells for the highest PC1 value genes, we observed a large overlap between genes in this dataset and those that are targets of the transcription factor Rpn4 (S3C Fig, highlighted in red).

### The transcription factor Rpn4 is involved in misfolded membrane protein stress

Rpn4 is a transcription factor that up-regulates genes with a proteasome-associated control element (PACE) in their promoters [53]. From our RNA-seq data, there was a remarkably high overlap between the genes that were observed to be up-regulated in *dfm1Δ* cells expressing Hmg2 and those that are known Rpn4 targets [53]. We reasoned that Rpn4 may be involved in adapting cells to misfolded membrane protein stress and predicted *rpn4Δ* cells should phenocopy *dfm1Δ* cells by exhibiting a growth defect induced by ERAD membrane substrates. Using the substrate-toxicity assay, we found expression of misfolded membrane proteins in *rpn4Δ* cells resulted in a growth defect equivalent to that seen in *dfm1Δ* cells (Figs 6A and S4A), indicating that Rpn4 is also required for alleviating misfolded membrane protein stress. As with *dfm1Δ* cells, this effect was specific to membrane protein expression, as expression of CPY* in *rpn4Δ* cells did not result in a growth defect (S4B Fig). This is in line with previous research demonstrating Rpn4 is activated in response to misfolded membrane protein accumulation and that misfolded membrane protein expression can result in proteasome impairment, even in WT cells [54,55]. Finally, we tested a transcription factor that can regulate Rpn4 and has many overlapping transcriptional targets with Rpn4, Pdr1 [56], and did not observe any growth defect in *pdr1Δ*+Hmg2 cells (S4C Fig).

As a readout for Rpn4 activity, we measured the abundance of a GFP-tagged version of the proteasome component Pre6-GFP in *dfm1Δ*+Hmg2 cells 0- and 5-hours after galactose induction through flow cytometry. Pre6 is a component of the 20S core of the proteasome that can be transcriptionally up-regulated by Rpn4, and Pre6-GFP has been used by others as a marker for the proteasome [57,58]. In comparison to WT control strains and other substrates tested, *dfm1Δ*+Hmg2 had a significant increase in Pre6-GFP after 5 hours (Fig 6B and 6C).

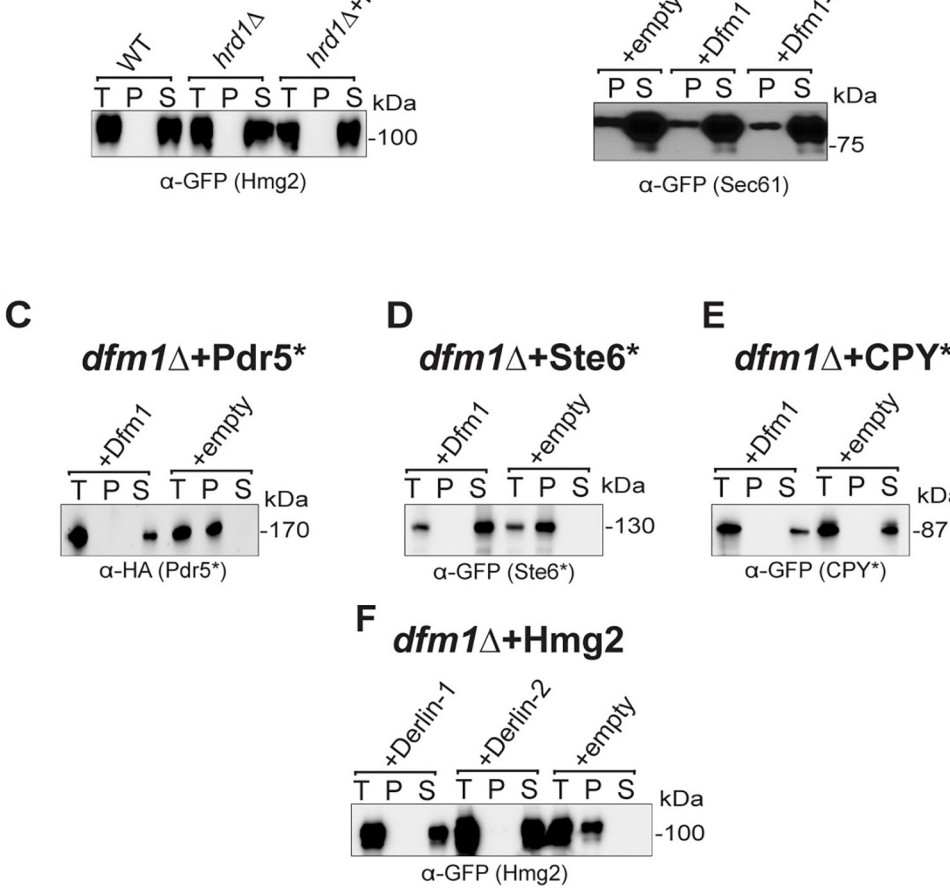

**Fig 4. Dfm1 specifically influences solubility of misfolded membrane proteins. (A)** Western blot of aggregated versus non-aggregated membrane proteins at the ER. Lysates from WT, *hrd1Δ*, or *hrd1Δ*+HRD1 cells containing HMG2-GFP were blotted using anti-GFP to detect Hmg2. T is total protein, P is pellet (ER aggregated) fraction, and S is soluble (ER non-aggregated) fraction. **(B)** Western blot of aggregated versus non-aggregated membrane proteins at the ER as in (A) except with *dfm1Δ* cells containing SEC61-GFP with add-back of EV, WT DFM1-HA, or DFM1-5Ashp-HA. Anti-GFP was used to detect SEC61-GFP. **(C)** Western blot of aggregated versus non-aggregated membrane proteins at the ER as in (A) except with *dfm1Δ* cells containing PDR5*-HA with add-back of WT DFM1-HA or EV. Anti-HA was used to detect PDR5*-HA. **(D)** Western blot of aggregated versus non-aggregated membrane proteins at the ER as in (A) except with *dfm1Δ* cells containing STE6*-GFP with add-back of WT DFM1-HA or EV. Anti-GFP was used to detect STE6*-GFP. **(E)** Western blot of aggregated versus non-aggregated membrane proteins at the ER as in (A) except with *dfm1Δ* cells containing CPY*-GFP with add-back of EV or WT DFM1-HA. Anti-GFP was used to detect CPY*-GFP. **(F)** Western blot of aggregated versus non-aggregated membrane proteins at the ER as in (A) except with *dfm1Δ* cells containing HMG2-GFP with add-back of EV, DERLIN-1-Myc, and DERLIN-2-Myc. Anti-Myc was used to detect DERLIN-1-Myc and DERLIN-2-Myc. Data information: All detergent solubility assays were performed with 3 biological replicates (*N* = 3). ER, endoplasmic reticulum; EV, empty vector.

## Misfolded membrane protein stress in dfm1Δ cells leads to proteasome impairment

Because Rpn4 appears to be active in membrane protein-stressed *dfm1Δ* cells, we hypothesized that proteasome function is impacted in *dfm1Δ* cells expressing an integral membrane protein. We tested this using an MG132 sensitivity assay developed by the Michaelis lab [54]. MG132 is a drug that reversibly inhibits proteasome function [59]. For this assay, cells in liquid culture

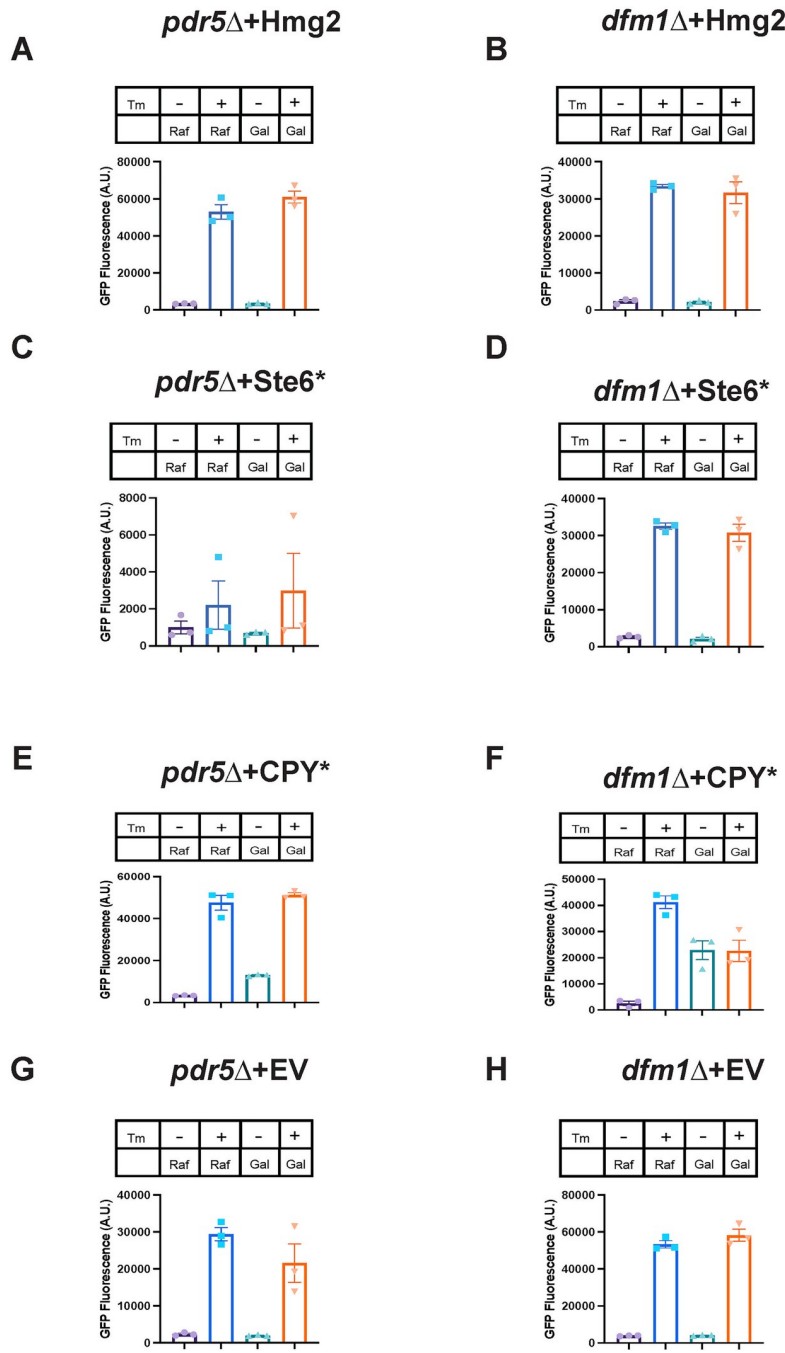

**Fig 5. Misfolded membrane protein stress in *dfm1Δ* cells does not activate the UPR. (A)** UPR activation for indicated strains with overexpression of a misfolded integral membrane protein. *pdr5Δ* cells containing GAL_pr-Hmg2-6Myc and 4xUPRE-GFP (a reporter that expresses GFP with activation of the UPR) were measured for GFP expression using flow cytometry every hour for 5 hours starting at the point of galactose induction and tunicamycin or equivalent volume of DMSO was added at the 1-hour time point. Figure depicts the GFP fluorescence in A.U. for indicated conditions 5 hours post-galactose addition. In figure legend, "Gal" indicates addition of 0.2% galactose to cultures and "Raf" indicates addition of 0.2% raffinose to culture, and "Tm" indicates presence (+) or absence (-) of 2 μg/mL tunicamycin. **(B)** Flow cytometry-based UPR activation assay as described in (A) except using *dfm1Δ* cells. **(C, E, and G)** Flow cytometry-based UPR activation assay as described in (A) except using cells containing GAL_pr-Ste6*-GFP, GAL_pr-CPY*-HA, or EV, respectively. **(D, F, and H)** Flow cytometry-based UPR activation assay as described in (B) except using cells containing GAL_pr-Ste6*-GFP, GAL_pr-CPY*-HA, or EV, respectively. Data information: All data are measured mean ± SEM; *N* = 3 biological replicates. The data underlying this figure can be found in Table A–H in S1 Data (Sheet 1). A.U., arbitrary unit; EV, empty vector; UPR, unfolded protein response.

were treated with MG132, plated, and counted the number of colony-forming units (CFUs) resulting from each strain. Due to the risk of the retrotranslocation defect being suppressed in *dfm1Δ* cells with constitutive expression of a misfolded membrane protein, and thus possibly artificially increasing the number of CFUs resulting from treatment of *dfm1Δ* cells with MG132, we opted to instead test *dfm1Δ hrd1Δpdr5Δ* cells. These cells are unable to suppress the retro-translocation defect of *dfm1Δ* cells, due to the absence of Hrd1, which has been characterized to function as an alternative retrotranslocon for membrane substrates when Dfm1 is absent [33]. We utilized the engineered misfolded membrane protein SUS-GFP as the substrate for these experiments. SUS-GFP contains the RING domain of Hrd1 and catalyzes its own ubiquitina-tion, thus still causing the stress that is elicited by ubiquitinated misfolded membrane proteins in *dfm1Δ* cells [60]. We predicted that cells with compromised proteasome function would be more sensitive to MG132 treatment, resulting in fewer CFUs. Strikingly, no CFUs resulted from MG132 treated *dfm1Δhrd1Δpdr5Δ* cells constitutively expressing SUS-GFP (Fig 6F). All others strains and treatments tested did not show as dramatic of a change in the number of CFUs, either with MG132 or DMSO treatment (Fig 6D–6F). These results demonstrate that protea-some function is impacted in *dfm1Δ* cells with misfolded membrane protein accumulation.

## Misfolded membrane protein stress does not cause proteasome sequestration

One hypothesis that we explored to understand the mechanism by which proteasomes are impaired with misfolded membrane protein stress was direct sequestration of proteasomes at the ER. Using *dfm1Δ* cells expressing EV or Hmg2, we used western blotting to detect ER recruitment of Pre6, a proteasome component (Fig 6G). Proteasome recruitment was similar between both strains. We also tested aggregation versus solubility of Pre6 at the ER in both strains and this was also not affected in either strain (Fig 6G). These results indicate an indirect mechanism for proteasome impairment in membrane protein-stressed cells.

## Growth defect in dfm1Δ cells is ubiquitination dependent

The observation that a growth defect triggered by misfolded membrane proteins is only seen in the absence of Dfm1, and not in cells lacking either of the ER E3 ligases Hrd1 and Doa10, led us to hypothesize that this growth defect is dependent upon ubiquitination of the misfolded membrane proteins. The substrate-toxicity assay results using *cdc48-2* cells indicate that the growth defect is not solely due to defective ERAD or the accumulation of ubiquitinated mis-folded membrane proteins. Nonetheless, we still explored the possibility that misfolded mem-brane protein-induced toxicity is dependent on substrate ubiquitination.

We examined whether growth defects were seen in either *dfm1Δhrd1Δ* or *dfm1Δdoa10Δ* cells expressing either Hmg2 (an Hrd1 target) or Ste6* (a Doa10 target), respectively (Fig 7A). These results showed no growth defect in the double mutants for which the membrane protein expressed was not ubiquitinated by the absent E3 ligase: *dfm1Δhrd1Δ* cells expressing Hmg2 and *dfm1Δdoa10Δ* cells expressing Ste6* (Fig 7A). We validated that substrates were indeed not ubiquitinated in E3 ligase knockouts via western blot for ubiquitin (Fig 7B). In contrast, a growth defect was observed in the double mutants for which the absent E3 ligase did not par-ticipate in ubiquitination of the expressed membrane protein: *dfm1Δhrd1Δ* expressing Ste6* and *dfm1Δdoa10Δ* expressing Hmg2. This indicates that growth stress in *dfm1Δ* cells is depen-dent upon ubiquitination of the accumulated misfolded membrane protein.

As an alternative approach to determine if membrane proteins must be ubiquitinated to cause toxicity in the absence of Dfm1, we tested the expression of well-characterized, stabilized Hmg2 mutants. These mutants, Hmg2 (K6R), Hmg2 (K357R), and Hmg2 (K6R, K357R), were

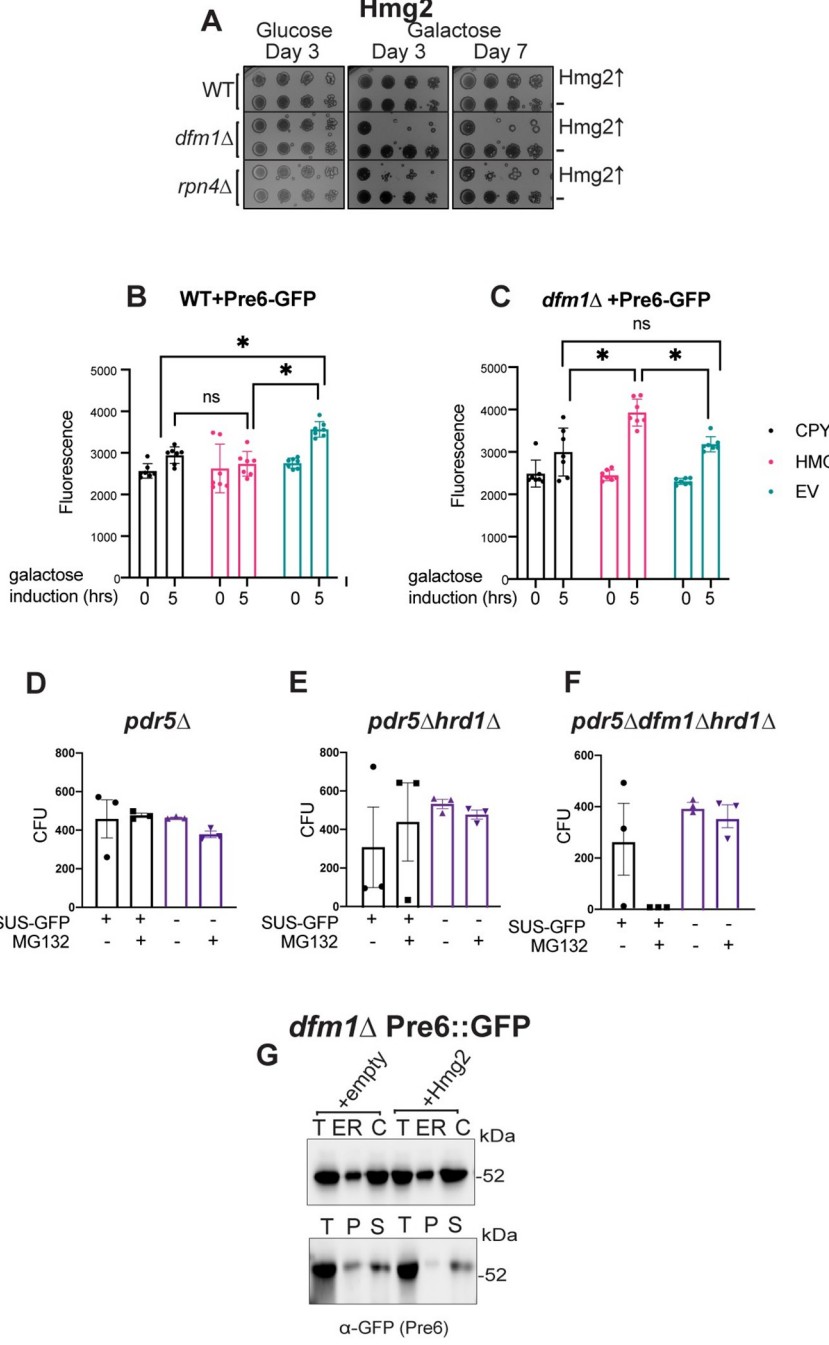

**Fig 6. Misfolded membrane protein toxicity results in proteasome impairment.** (**A**) WT, *dfm1Δ*, and *rpn4Δ* cells containing either GAL_pr_-HMG2-GFP or EV were compared for growth by dilution assay. Each strain was spotted 5-fold dilutions on glucose or galactose-containing plates to drive Hmg2-GFP overexpression, and plates were incubated at 30°C. Three biological replicates and 2 technical replicates (*N* = 3). (**B**) PRE6-GFP levels as measured by flow cytometry at 0 versus 5 hours post-galactose induction in WT cells containing either EV, GAL_pr_-CPY*-HA, or GAL_pr_-HMG2-GFP. (**C**) Pre6-GFP levels as in (B) except in *dfm1Δ*. (**D**) Quantification of CFUs formed on appropriate selection plates from proteasome sensitivity inhibition assay. *pdr5Δ* cells containing SUS-GFP or EV in log phase were treated with 25 uM of proteasome inhibitor MG132 or equivalent volume of DMSO for 8 hours and samples were diluted 1:500 and 50 uL of each sample was plated. (**E**) Proteasome sensitivity assay as in (D) except using *hrd1Δpdr5Δ* cells. (**F**) Proteasome sensitivity assay as in (D) except using *dfm1Δhrd1Δpdr5Δ* cells. Data information: For (B) and (C), all data are mean ± SEM, with 7 biological replicates (*N* = 7). For (D), (E), and (F), all data are mean ± SEM, 3 biological replicates and 2 technical replicates (*N* = 3); statistical significance is displayed as two-tailed unpaired *t* test, *\*P* < 0.05, ns, not significant. (**G**) Western blot of Pre6 in cytosol versus ER (top panel) and

aggregated (pelleted) versus non-aggregated (soluble) Pre6 at the ER (bottom panel). Lysates from *dfm1Δ*, cells containing Pre6-GFP and HMG2-6Myc or EV were blotted using anti-GFP to detect Pre6. T is total protein, ER is endoplasmic reticulum protein fraction, C is cystolosic protein fraction, P is pellet (ER aggregated) fraction, and S is soluble (ER non-aggregated) fraction. The data underlying this figure can be found in Table I–K in S1 Data (Sheet 2). CFU, colony-forming unit; EV, empty vector.

previously identified by the Hampton lab in a genetic screen for stabilized Hmg2 mutants [61]. Both K➞R stabilized mutations disrupt Hmg2 ubiquitination, and these sites are hypothesized to be Hmg2 ubiquitination sites. While the Hampton lab has shown that ubiquitination levels of both substrates are nearly undetectable with western blot, they also showed that the K6R mutant is not further stabilized in an ERAD deficient background, while the K357R mutant is slightly more stable in an ERAD deficient background than in a WT background [61]. We propose that because of this slight level of degradation in the K357R mutant, some fraction of this mutant must be ubiquitinated and targeted to the Hrd1 ERAD pathway. Our model predicts that toxicity of misfolded membrane proteins is ubiquitination dependent. Thus, we would expect that the fully stabilized Hmg2-K6R with negligible ubiquitination should not elicit a growth defect, whereas Hmg2-K357R, which is a poor ERAD substrate but still can still be targeted for degradation, should elicit a growth defect in *dfm1Δ* cells. Indeed, we observed no growth defect in *dfm1Δ* cells expressing the K6R mutant, while the K357R mutant still showed a growth defect. Moreover, the growth defect is still observed in the double mutant Hmg2-(K6R, K357R), which phenocopies Hmg2-K6R, in that it is completely stabilized, consistent with the model that growth stress in the absence of Dfm1 is dependent on the accumulation of ubiquitinated membrane substrates (Fig 7C).

## Ubiquitin homeostasis is disrupted with misfolded membrane protein accumulation

There is increasing evidence that suggests ubiquitin homeostasis and maintenance of the free ubiquitin pool is critical for cellular survival under normal and stress conditions [62–65]. Because we observed that growth defect in *dfm1Δ* cells is dependent on ubiquitination of membrane substrates, we hypothesized that ubiquitin conjugation to accumulating membrane proteins reduces the availability of free ubiquitin, impacting cell viability.

One hypothesis that would explain substrate ubiquitination dependency of the growth defect in *dfm1Δ* cells is that the pool of monomeric ubiquitin is depleted by accumulation of misfolded membrane proteins. If this hypothesis is correct, exogenous ubiquitin should rescue the growth defect seen from substrate-induced stress in *dfm1Δ* cells. To that end, *dfm1Δ* +Hmg2 cells harboring a plasmid containing ubiquitin under the control of the copper inducible promoter CUP1 [66] were tested in the substrate-toxicity assay. These cells were plated on 2% galactose and 50 μM copper to induce expression of Hmg2 and ubiquitin in *dfm1Δ* cells, respectively. Notably, supplementation of ubiquitin restored the growth defect (Fig 7D). We blotted for monomeric ubiquitin to determine whether this pool is depleted in *dfm1Δ*+Hmg2 cells and found that it was reduced compared to WT and *hrd1Δ* strains with Hmg2 (Fig 7E and 7F). In contrast, *dfm1Δ* without overexpressed Hmg2 do not show a decrease in monomeric ubiquitin (S5A and S5B Fig).

## Deubiquitinases prevent or resolve misfolded membrane protein stress

The deubiquitinase (DUB) Ubp6 is a peripheral subunit of the proteasome and recycles ubiquitin from substrates prior to proteasome degradation [65]. Accordingly, *ubp6Δ* cells were employed in the substrate-toxicity assay to determine whether this protein is involved in

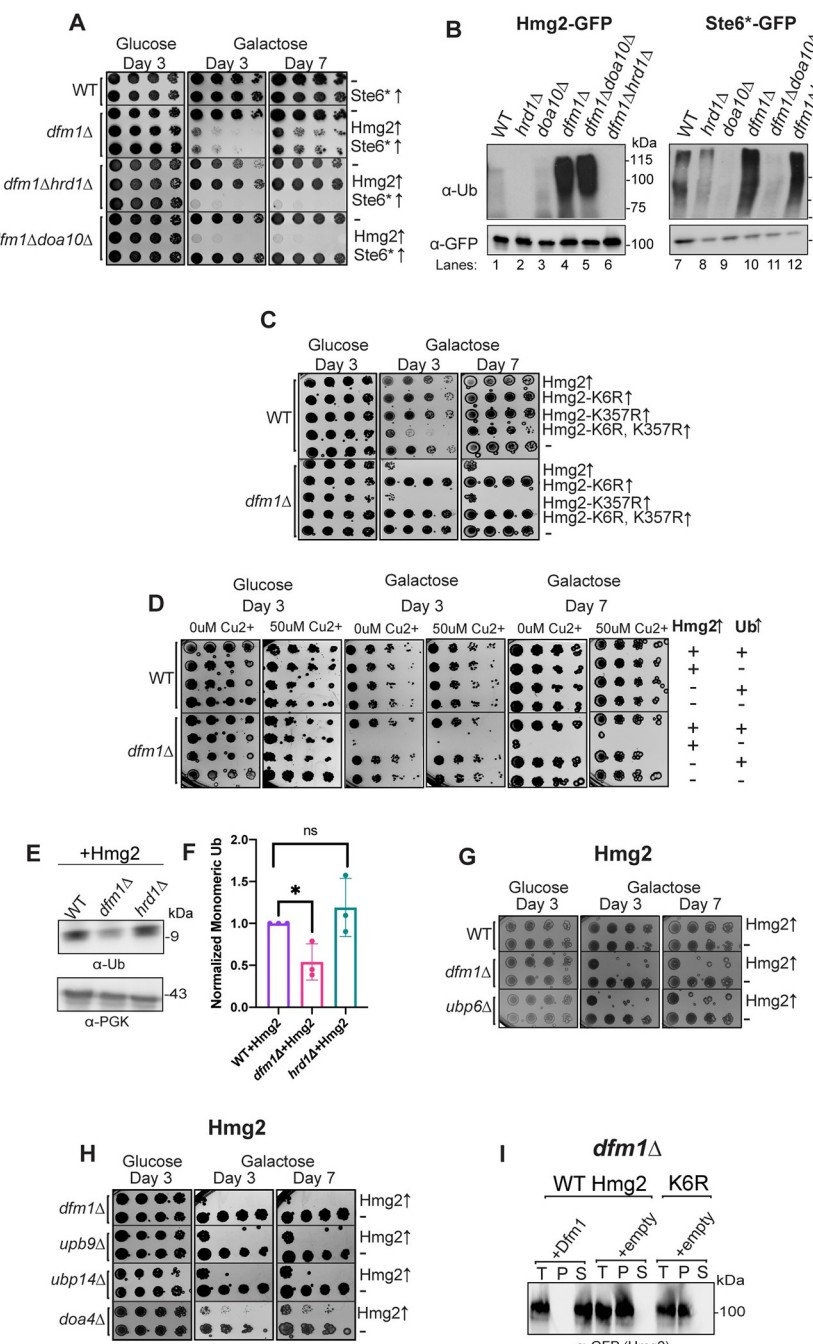

**Fig 7. Ubiquitin stress contributes to misfolded membrane protein toxicity. (A)** WT, *dfm1Δ*, *dfm1Δhrd1Δ*, and *dfm1Δdoa10Δ* cells containing either GAL_pr-Hmg2-GFP, GAL_pr-STE6*-GFP, or EV were compared for growth by dilution assay. Each strain was spotted 5-fold dilutions on glucose or galactose-containing plates to drive Hmg2-GFP overexpression, and plates were incubated at 30°C. **(B)** Indicated strains expressing either Hmg2-GFP or Ste6*-GFP were grown to log-phase, lysed, and microsomes were collected and immunoprecipitated with α-GFP conjugated to agarose beads. Samples were then subjected to SDS-PAGE and immunoblotted by α-Ubiquitin and α-GFP. Three biological replicates (*N* = 3). **(C)** Dilution assay as described in (A) except using WT and *dfm1Δ* cells containing either GAL_pr-Hmg2-GFP, GAL_pr-Hmg2-K6R-GFP, GAL_pr-Hmg2-K357R-GFP, GAL_pr-Hmg2- (K6R and K357R)-GFP, or EV. **(D)** WT and *dfm1Δ* cells containing either CUP1_pr-Ub or EV and GAL_pr-HMG2-GFP or EV were compared for growth by dilution assay. Each strain was spotted 5-fold dilutions on glucose or galactose-containing plates to drive Hmg2-GFP overexpression, and plates were incubated at 30°C. Galactose plates containing 50 μM Cu2+ were used to allow expression of Ub driven by the CUP1 promoter. **(E)** Western blot of monomeric ubiquitin in WT, *dfm1Δ*, and *hrd1Δ* expressing HMG2-GFP. Anti-ubiquitin was used to blot for ubiquitin and anti-PGK1 was used to blot for PGK1

as a loading control. **(F)** Quantification of western blots from (E). Each strain was normalized to PGK1 and the monomeric ubiquitin quantification of WT+HMG2-GFP was used to normalize all strains. **(G)** Dilution assay as described in (A) *dfm1Δ*, *ubp9Δ*, *ubp14Δ*, and *doa4Δ* cells. **(H)** Dilution assay as described in (A) except using WT, *dfm1Δ*, and *ubp6Δ* cells containing either GAL$_{pr}$-HMG2-GFP or EV. **(I)** Western blot of aggregated versus soluble membrane proteins at the ER. Lysates from *dfm1Δ* cells containing HMG2-GFP or HMG2-K6R-GFP with EV or DFM1-HA were blotted using anti-GFP to detect Hmg2. T is total fraction, P is pellet (ER aggregated) fraction, and S is soluble (ER non-aggregated) fraction. Data information: All dilution growth assays were performed in 3 biological replicates and 2 technical replicates ($N = 3$). For (F), all data are mean ± SEM, 3 biological replicates ($N = 3$); statistical significance is displayed as two-tailed unpaired *t* test, $^*P < 0.05$, ns, not significant. Detergent solubility assay in (H) was performed with 3 biological replicates ($N = 3$). The data underlying this figure can be found in Table L in S1 Data (Sheet 3). ER, endoplasmic reticulum; EV, empty vector.

alleviating misfolded membrane protein stress by replenishing the free ubiquitin pool. By utilizing the substrate-toxicity assay, we found Hmg2 or Ste6* expression causes a growth defect in *ubp6Δ* cells (Figs 7F and S5C). Like *dfm1Δ* and *rpn4Δ* cells, this growth defect was specific to misfolded membrane proteins and was not observed with CPY* (S6D Fig). To confirm whether this effect was specific to Ubp6, we also tested DUB Doa4, another regulator of free ubiquitin, in the substrate-toxicity assay. Unexpectedly, we found that *doa4Δ* cells phenocopy *ubp6Δ* cells with Hmg2 expression (S6A Fig). From this observation, we tested a collection of DUB KOs in the substrate-toxicity assay. Of the 14 yeast DUBs tested (out of 22 DUBs total), we observed a growth defect with both *ubp9Δ* and *ubp14Δ* cells (Figs 7H and S6B). Interestingly, Ubp6, Doa4, and Ubp14 have all previously been implicated in ubiquitin homeostasis and, to date, no research has been conducted into the specific role of Ubp9 [67]. Interestingly, when a misfolded cytosolic substrate, ΔssCPY*, is expressed in *ubp6Δ*, *doa4Δ*, *ubp9Δ*, and *ubp14Δ* cells, there is no growth defect observed (S6C Fig).

## Absence of deubiquitinases and RPN4 in combination with DFM1 do not exacerbate toxicity

We tested double knockouts of *dfm1Δrpn4Δ*, *dfm1Δubp6Δ*, and *rpn4Δubp6Δ* cells expressing Hmg2 in the substrate-toxicity assay to determine whether these genetic backgrounds display the same or different growth defect than either of the single knockouts. Expression of either Hmg2 or Ste6* in either *dfm1Δrpn4Δ* orF020*dfm1Δubp6Δ* cells resulted in a growth defect that phenocopied that observed in any of the single knockouts (S7A and S7B Fig), whereas expression of CPY* showed no growth defect (S7C Fig). In contrast, *rpn4Δubp6Δ* cells showed a growth defect in the absence of substrates, whereas *rpn4Δ* and *ubp6Δ* displayed normal growth. Moreover, *rpn4Δubp6Δ* cells along with expression of Hmg2 or Ste6* resulted in synthetic lethality (S7A–S7C Fig). This indicates that there is an exacerbation of stress in *rpn4Δubp6Δ* background, whereas there is no increase in toxicity when RPN4 or UBP6 are knocked out in combination with DFM1. It is likely that there are several parallel pathways contributing to preventing stress from misfolded membrane proteins and resolving this stress, and Dfm1 appears to be one of the major mediators of misfolded membrane stress prevention.

We also tested expression of previously described Hmg2 mutants K6R and K357R in *rpn4Δ* and *ubp6Δ* cells (S7D Fig). As with *dfm1Δ* cells expressing these mutants, expression of Hmg2-K6R does not cause toxicity while Hmg2-K357R does cause toxicity in both *rpn4Δ* and *ubp6Δ*. Thus, ubiquitination of misfolded membrane proteins influences toxicity in *dfm1Δ*, *rpn4Δ*, and *ubp6Δ* cells.

## Misfolded protein aggregation toxicity requires protein ubiquitination

We originally hypothesize that ubiquitination of membrane proteins was promoting those proteins to become aggregated. To test this hypothesis, we measured aggregation versus

solubility of Hmg2-K6R in *dfm1Δ* cells in the detergent solubility assay (Fig 7I). Surprisingly, Hmg2-K6R phenocopied Hmg2 in *dfm1Δ* cells, with virtually all of the protein being in the aggregated fraction. This demonstrates ubiquitin does not influence aggregation and not all misfolded membrane protein aggregates are toxic.

### Increased expression of Dfm1 relieves misfolded membrane protein stress in rpn4Δ and ubp6Δ cells

Using the substrate-toxicity assay, we examined whether increasing expression of Dfm1 could relieve growth stress in *rpn4Δ* and *ubp6Δ* cells expressing Hmg2. We utilized the substrate-toxicity assay with the addition of galactose inducible Dfm1 to address this question. Increasing Dfm1 in both *rpn4Δ*+Hmg2 and *ubp6Δ*+Hmg2 cells restored normal growth (S7E Fig). Importantly, endogenous Dfm1 is already present in these cells, but increasing expression level relieves toxicity caused by misfolded membrane proteins.

## Discussion

Proper protein folding and efficient elimination of misfolded proteins is imperative for maintaining cellular health. Accumulation of misfolded proteins, which is a widespread phenomenon in aging and diseased cells, is deleterious to cells and can impact cellular function. Despite membrane proteins accounting for one-quarter of proteins in the cell, there is a dearth of research into the mechanisms cells use to prevent misfolded membrane protein toxicity. In this study, we sought to understand how cells prevent toxicity from misfolded proteins and how they are impacted by misfolded membrane protein stress. By employing our genetically tractable substrate-toxicity assay, we found that the source of cellular toxicity was aggregation of ubiquitinated misfolded membrane proteins and that Dfm1's rhomboid motifs, lipid thinning function, and substrate engagement function are required for its chaperone-like function towards aggregation-prone membrane substrates. We propose a model in which ubiquitinated misfolded membrane proteins in *dfm1Δ* cells form aggregates, resulting in a reduction in monomeric ubiquitin and compromised proteasome function. Overall, our studies unveil a new role for rhomboid pseudoproteases in mitigating the stress state caused by ERAD membrane substrates, a function that is independent of their retrotranslocation function.

Our results above (Fig 2B–2D) indicate differential requirements for Dfm1's role in membrane substrate retrotranslocation, versus its role in stress alleviation. These results are fascinating, because of all the retrotranslocation-deficient mutants tested, we were able to identify a mutant that was still able to rescue the growth defect observed in *dfm1Δ*+Hmg2 cell. This indicates a bifurcated role of Dfm1 in retrotranslocation and membrane protein stress alleviation. The retrotranslocation defective mutants that did not restore growth were mutations of conserved rhomboid protein motifs (WR and Gx3G), mutants that obliterate substrate engagement (Loop 1 mutants: F58S, L64V, and K67E), and mutants that reduce the ability of Dfm1 to distort the ER membrane (TMD 2 mutants: Q101R and F107S). This indicates the substrate binding and lipid distortion roles of Dfm1 that are imperative for retrotranslocation are also imperative for alleviation of misfolded membrane protein stress. In contrast, the SHP box mutant, which prevents Cdc48 binding to Dfm1, restores growth in *dfm1Δ*+Hmg2 cells (Fig 2D). While Cdc48 binding to Dfm1 is critical for retrotranslocation, this is not a requirement for Dfm1's role in preventing membrane proteotoxicity. Previous work from our lab indicates transient interactions between membrane substrates and Dfm1 still occurs even when Dfm1's Cdc48 recruitment activity is impaired [19]. This suggests that this level of physical interaction is sufficient for Dfm1 to directly act on substrates to prevent membrane substrate-induced stress.

Our results from both this study and previous work from the lab on Dfm1's function indicate Dfm1 acts directly on misfolded membrane proteins to promote influence aggregation [19]. Firstly, all of the L1 mutants of Dfm1, which have previously been shown to ablate binding of Dfm1 to misfolded membrane proteins, such as Hmg2, are not able to restore non-aggregated Hmg2. Secondly, we demonstrate here that DDM solubilized (non-aggregated) Hmg2 still interacts with Dfm1, as shown through co-immunoprecipitation (Fig 3C). Lastly, both human Derlin-1 and Derlin-2 are able to restore non-aggregated Hmg2 in *dfm1Δ* cells (Fig 4F). It seems unlikely that if Dfm1 was influencing solubility of Hmg2 through an indirect route that mammalian derlins, which have diverged significantly from Dfm1, would still influence aggregation.

We previously demonstrated that expression of integral membrane ERAD substrate induces toxicity in yeast cells when Dfm1's function is impaired. Remarkably, this strong growth defect phenotype is unique to *dfm1Δ* strains: other equally strong ERAD deficient mutants, both upstream or downstream of Dfm1 (*hrd1Δ* or *cdc48-2*), show no growth stress upon similar elevation of ERAD integral membrane substrates. Thus, the growth effects above suggest the intriguing possibility that Dfm1 has a unique role in this novel ER stress.

Our data on the ability of Dfm1 to influence misfolded membrane protein aggregation provides evidence that this is the mechanism by which Dfm1 prevents misfolded membrane protein toxicity. We find that both WT Dfm1 and Dfm1-5Ashp expression result in non-aggregated Hmg2 (Fig 3A and 3B). In contrast, the retrotranslocation defective Dfm1 rhomboid motif mutants, L1 mutants, and TMD2 mutants are not able to influence Hmg2 aggregation. This is in agreement with our observation that both WT Dfm1 and Dfm1-5Ashp can restore normal growth in *dfm1Δ* cells in the S-T assay, but the rhomboid motifs mutants cannot (Fig 2B and 2D). The exact mechanism by which Dfm1 results in non-aggregated Hmg2 is unclear. We propose 2 possible models that will be important to distinguish between in future works. In one model, Dfm1 functions as a disaggregase to physically separate misfolded membrane proteins from existing protein aggregates. In another model, Dfm1 functions as a holdase to maintain misfolded membrane proteins in a non-aggregated state. We believe present data indicates a holdase function is more likely, as Dfm1 interacts with solubilized Hmg2 (Fig 3C), potentially preventing it from forming aggregates, but this hypothesis will need to be tested more mechanistically in future studies. While the ability of Dfm1-5Ashp to influence Hmg2 aggregation in *dfm1Δ* cells indicates that Dfm1's chaperone-like ability is likely ATP-independent, we cannot exclude the possibility that Dfm1 recruits another ATPase besides Cdc48, independent of the SHP box motif. Another possibility is that Dfm1 itself can bind and hydrolyze ATP. There are a growing number of identified ATP-independent disaggregases [68], including 1 membrane protein dissagregase identified in plants [69]. Understanding how Dfm1 influences the aggregation of membrane substrates will be an important future line of inquiry.

By analyzing the transcriptome upon triggering this unique membrane substrate-induced stress state, we find that many proteasomal subunits are up-regulated. Interestingly, Rpn4—a transcription factor known to induce proteasome subunit expression—up-regulates many of the proteasomal subunits up-regulated in our transcriptome analysis. One interpretation of our data is that accumulation of integral membrane proteins results in reduced proteasome efficiency, which triggers Rpn4-mediated up-regulation of proteasome subunits. Indeed, we and others have shown that *rpn4Δ* cells phenocopy *dfm1Δ* cells by exhibiting a growth defect upon expression of ERAD integral membrane substrates and not ERAD lumenal substrates [54]. This was also supported by our above studies showing ERAD membrane substrates exacerbate cellular growth defects when proteasome function is compromised with treatment of proteasome inhibitor, MG132 (Fig 6D–6F). These data indicate that cells require optimal

proteasome activity to avoid the proteotoxicity associated with integral membrane ERAD substrates.

The facile and genetically tractable substrate-toxicity assay allowed us to ascertain how membrane substrates cause the growth defect phenotype when Dfm1 is absent. Intriguingly, no growth defect was observed in *dfm1Δ* cells expressing the K6R Hmg2 mutant (with negligible ubiquitination), while the K357R Hmg2 mutant (with slight ubiquitination) still showed a growth defect, suggesting the source of Dfm1-mitigated stress is ubiquitination of the substrates. We reasoned that accumulation of ubiquitinated ERAD membrane substrates disrupts the ubiquitin pool through excessive ubiquitination of substrates and concomitant reduction of the ubiquitin pool. Indeed, a collection of DUB mutants (*ubp6Δ*, *doa4Δ*, *ubp14Δ*)—known for their role in replenishing the ubiquitin pool through their deubiquitinating function—is unable to mitigate the proteotoxic effect of integral membrane substrates, and proteotoxic stress is rescued with exogenous addition of ubiquitin molecules in *dfm1Δ*+Hmg2 cells (Fig 7D, 7G and 7H). This observation is extended in mammalian studies in which a mouse line with a loss-of-function mutation in Usp14, the mammalian homolog of Ubp6, reduces the free ubiquitin pool in neurons and results in ataxia that can be rescued with exogenous ubiquitin expression [63]. The reduction we observed in monomeric ubiquitin in *dfm1Δ*+Hmg2 cells was approximately half of that observed in WT+Hmg2 cells (Fig 6E). The hypothesis that this reduction is enough to contribute to toxicity in these cells is supported both by our experiment demonstrating the exogenous ubiquitin restores growth in the substrate-toxicity assay (Fig 7D) and by the observation that ataxic Usp14-deficient mice only show about a 25% reduction in monomeric ubiquitin in most tissues [63]. Perhaps what is most fascinating is that the stress state is only induced by excessive ubiquitination of integral membrane substrates and not soluble proteins residing in the cytosol (S6C Fig), suggesting the source of stress is due to excessive ubiquitination of substrates at the ER membrane.

There is an emerging body of evidence that protein aggregation is not inherently toxic [70–72]. We propose a model in which 2 conditions need to be met for misfolded membrane protein accumulation to become toxic; (1) the misfolded membrane proteins must become aggregated; and (2) the misfolded membrane proteins must be ubiquitinated (Fig 8). If only one of these conditions is met, there is no toxicity observed in the substrate-toxicity assay. For example, Dfm1-5Ashp restores growth and results in non-aggregated Hmg2, even without restoring retrotranslocation (Figs 2E and 3A). These accumulated membrane proteins would still be expected to be ubiquitinated, but no toxicity is observed without aggregated Hmg2 in this circumstance. Conversely, the nonubiquitinated Hmg2-K6R does not cause toxicity, even though virtually all the protein is aggregated in *dfm1Δ* cells (Fig 7I). Additionally, the ability to influence aggregation of misfolded membrane proteins appears specific to Dfm1 among ERAD machinery, as *cdc48-2* cells with an overexpressed misfolded membrane protein do not display growth stress, and nearly all Hmg2 non-aggregated in *hrd1Δ* cells in the detergent solubility assay (Figs 1E, 1F and 4A).

Molecular chaperones have long been identified for their role in protein quality control systems, including ERAD, for their ability to triage terminally misfolded proteins to degradation machinery. In recent years, more studies have shown a dual function of protein quality control machinery in directly controlling degradation and being chaperones [39,73]. We have now provided evidence for rhomboid pseudoproteases, a subclass of proteins widely recognized as involved in protein quality control, having chaperone-like function. This raises the question of whether chaperone ability is more widespread among other protein quality control components, specifically those known to bind to membrane proteins. Previous work from the Brodsky lab demonstrated that aggregation-prone ER proteins are more likely to be targeted by ERAD and are disaggregated by the ATP-dependent cytoplasmic disaggregase Hsp104, which

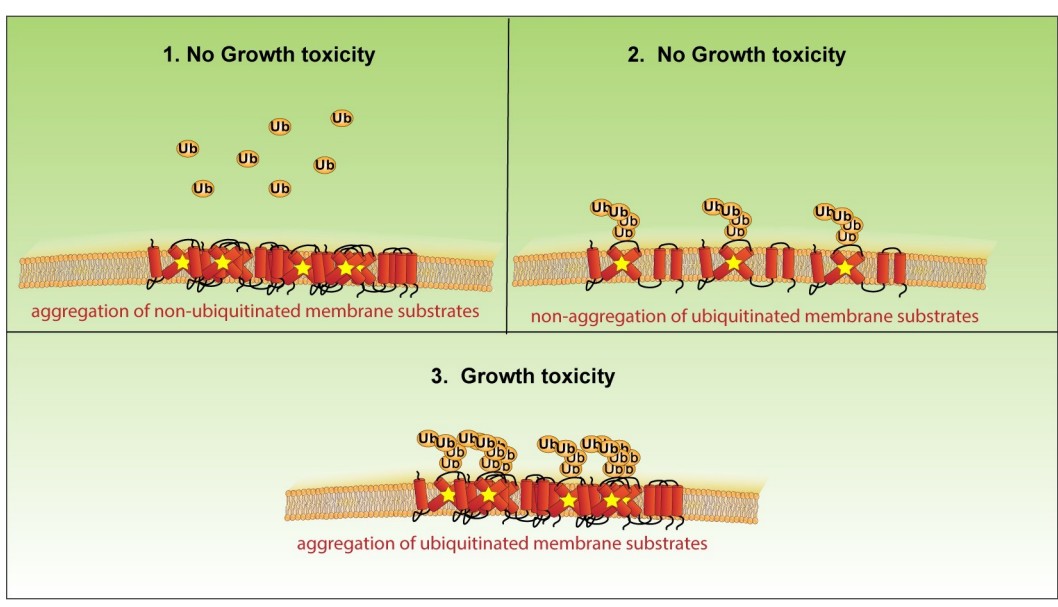

**Fig 8. Model for misfolded membrane protein-induced toxicity.** A model depicting how accumulation of ER-resident misfolded membrane proteins would induce growth toxicity. (1) No growth toxicity is observed when misfolded membrane proteins aggregate but are not ubiquitinated. (2) No growth toxicity is observed when misfolded membrane proteins are ubiquitinated, but not aggregated. (3) Growth toxicity is observed when misfolded membrane proteins are both ubiquitinated and aggregated. ER, endoplasmic reticulum.

aids in retrotranslocation [74]. Our results demonstrate that a component of membrane protein retrotranslocation machinery, Dfm1, also has a chaperone-like function to aid in retrotranslocation. The Carvalho group has demonstrated that the Asi complex involved in inner nuclear membrane protein quality control in yeast and the mammalian ERAD factor membralin are able to recognize TMDs of misfolded proteins [75,76]. It is possible that chaperone function has arisen more than once evolutionarily among proteins involved in membrane protein quality control.

Rhomboid pseudoproteases have been recognized for over a decade as being involved in a diverse array of cellular process, from protein quality control to cell signaling to adaptations to cellular stress [24,30,46,77,78]. Our lab and others have made progress towards understanding how these proteins are able to function is such diverse cellular process without an enzymatic function. With the knowledge that several derlin proteins are chaperone-like proteins, it will be of extreme interest to determine if this function is conserved among all rhomboid pseudoproteases, and even among the active rhomboid proteases. Two specific areas of interest include determining the conservation of this chaperone-like function and identifying the repertoire of substrates that can be solubilized by rhomboid pseudoproteases. There are 2 subclasses of rhomboid pseudoproteases, iRhoms and derlins [79]. Both of these classes are evolutionarily distinct and it will be of interest to determine if chaperone-like ability is only specific to derlins and not to iRhoms. Derlins are known to function in retrotranslocation of a wide variety of substrates, including disease-associated membrane substrates. In this study, we observed accumulation of both WT and the disease-causing CFTRΔF508 caused growth stress in *dfm1Δ* cells [27,32,45,46]. Surprisingly, we found that heterologous expression of both human Derlin-1 and Derlin-2 restores growth in yeast *dfm1Δ*+Hmg2 cell and results in non-aggregated Hmg2, implying the chaperone-like function is a conserved feature among derlin rhomboid pseudoproteases. Moreover, research from our lab demonstrated that Derlin-1 and

Derlin-2 do not support ERAD-M retrotranslocation in *dfm1Δ* cells [19]. This indicates that Derlin-1 and Derlin-2 relieve toxicity in *dfm1Δ*+Hmg2 cells, without restoring retrotranslocation, through a conserved chaperone-like function.

Our studies provide the first evidence that the derlin subclass of rhomboid pseudoproteases function as chaperone-like proteins by influencing aggregation of misfolded membrane substrates. Findings gleaned from our studies hold great promise for foundational and translational arenas of cell biology, since fundamental understanding of a membrane protein chaperone will aid in understanding a plethora of diseases associated with misfolded membrane proteins such as cystic fibrosis, retinal degeneration, and neurodegenerative diseases.

## Methods

### Plasmids and strains

Plasmids used in this study are listed in S1 Table. Plasmids for this work were generated using standard molecular biological cloning techniques via PCR of genes from yeast genomic DNA or plasmid followed by ligation into a specific restricted digested site within a construct and verified by sequencing (Eton Bioscience and Plasmidsaurus). Primer information is available upon request.

A complete list of yeast strains and their corresponding genotypes are listed in S2 Table. All strains used in this work were derived from S288C or Resgen. Yeast strains were transformed with DNA or PCR fragments using the standard LiOAc method in which null alleles were generated by using PCR to amplify a selection marker flanked by 30 base pairs of the 5′ and 3′ regions, which are immediately adjacent to the coding region of the gene to be deleted. The selectable markers used for making null alleles were genes encoding resistance to G418 or Clo-Nat/nourseothricin or HIS3. After transformation, strains with drug markers were plated onto YPD followed by replica-plating onto YPD plates containing 500 μg/mL G418 or 200 μg/mL nourseothricin, or minimal media (-His) plates. All gene deletions were confirmed by PCR.

### Galactose induction

For strains with plasmids containing galactose inducible promoters, protein expression was achieved by growing proteins overnight in appropriate selection media containing 2% raffinose as carbon source. The following day, samples were diluted between 0.10 and 0.20 OD at 600 nm (diluted absorbance was assay dependent). Cells in log phase were induced by adding 0.2% galactose to media. Minimum time requirement for robust protein expression was determined for strains using flow cytometry and was 2 or 3 hours for every strain used.

### Flow cytometry

Yeast were grown in minimal medium with 2% raffinose and 0.2% galactose and appropriate amino acids into log phase (OD600 < 0.2). The BD Biosciences FACS Calibur flow cytometer measured the individual fluorescence of 10,000 cells. Experiments were analyzed using Prism8 (GraphPad).

### Unfolded protein response activation assay

Strains were inoculated overnight in minimal media (-His) with 2% raffinose. The following day, samples were diluted to 0.20 OD in of minimal media (-His) and allowed to grow to log phase. Samples were then diluted to 0.30 OD before adding 20% galactose to a final concentration of 0.2% galactose or an equal volume of dH2O. Timer was started after galactose addition and samples were measured using flow cytometry, as described above, every hour, starting

from the 0-hour mark, and ending at the 5-hour mark. At the 1-hour time point, samples were treated with either 2 μg/mL tunicamycin or an equal volume of DMSO.

## Hac1 splicing PCR

Strains were prepared the same as for the UPR activation assay, except they were grown in minimal media (-Ura -His) with 2% raffinose. After 5 hours of incubation with 0.2% galactose and 2 μg/mL tunicamycin, samples were pelleted and washed with dH2O. RNA from samples was extracted using Qiagen RNeasy Mini Kit. Samples were ethanol precipitated by adding 1 uL of Glycoblue (Thermo Fisher), 50 uL of 7.5 M ammonium acetate, and 700 uL of chilled 100% ethanol. Tubes were then stored at −80˚C for between 3 hours to overnight. Samples were then centrifuged at $13,000 \times g$ for 30 minutes at 4˚C and supernatant was removed. Pellets were washed twice with 75% ethanol and centrifuged at room temperature at 13,000xg for 30 seconds. After drying the pellet, it was resuspended in 15 uL of molecular grade water. A total of 250 ng of RNA from each sample was used to generate cDNA using a standard protocol for ProtoscriptII Reverse Transcriptase (NEB), except with 1 uL of Oligo $(dT)_{12-18}$ (Thermo Fisher Scientific) used for primer. Wizard SV Gel and PCR Clean-Up System (ProMega) was used on cDNA samples. Hac1 mRNA was amplified using forward primer 5′ACTTGGCTATCCCTACCAACT 3′ and reverse primer 5′ATGAATTCAAACCTGACTGC 3′. PCR products were resolved on a 2% agarose gel.

## MG132 sensitivity assay

MG132 sensitivity assay was performed using a protocol adapted from [54]. In brief, cultures grown minimal media (-his) with 2% dextrose. Cultures in log phase were split and treated with either 50 μM MG132 in DMSO or an equal volume of DMSO alone and incubated for 8 hours at 30˚C. Cultures were diluted 1:500 and 100 μL of sample was plated onto minimal media (-His) plates and grown at 30˚C for 3 days. Two technical replicates and 3 biological replicates were done for each strain. CFUs were counted for using the ProMega Colony Counter application for iPhone.

## Spot dilution assay (substrate-toxicity assay)

Yeast strains were grown in minimal selection media (-His) supplemented with 2% dextrose to log phase (OD600 0.2 to 0.3) at 30˚C. Approximately 0.10 OD cells were pelleted and resuspended in 1 mL dH2O. A total of 250 μL of each sample was transferred to a 96-well plate where a 5-fold serial dilution in dH2O of each sample was performed to obtain a gradient of 0.1–0.0000064 OD cells. The $8 \times 6$ pinning apparatus was used to pin cells onto synthetic complete agar plates supplemented with 2% dextrose or 2% galactose. Plates were incubated at 30˚C and removed from the incubator for imaging after 3 days and again after 7 days. All experiments were done in biological triplicates with technical replicates.

## RNA sequencing

RNA was isolated using a Qiagen RNeasy kit using standard protocol for yeast. Samples were eluted twice with 30 μL of molecular grade water. To cleanup samples, 1 μL of DNase was added to each sample and was incubated at 37˚C for 25 minutes. Approximately 6 μL of DNase inactivation buffer was added to samples and was incubated for 2 minutes. Samples were spun down at 10,000xg for 1.5 minutes and supernatant was transferred to a new microfuge tube. Samples were ethanol precipitated by adding 1 μL of Glycoblue (Thermo Fisher), 50 μL of 7.5 M ammonium acetate, and 700 μL of chilled 100% ethanol. Tubes were then stored

at −80°C for between 3 hours to overnight. Samples were then centrifuged at 13,000 × *g* for 30 minutes at 4°C and supernatant was removed. Pellets were washed twice with 75% ethanol and centrifuged at room temperature at 13,000xg for 30 seconds. After drying the pellet, it was resuspended in 15 μL of molecular grade water. Samples were measured for RNA concentration and an equal concentration of each sample was measured out into a total of 50 μL of molecular grade water and RNA-seq was performed as previously described [80] or as follows. Poly A-enriched mRNA was fragmented, in 2× Superscript III Mg2+ containing first-strand buffer with 10 mM DTT (Invitrogen), by incubation at 94°C for 9 minutes, then immediately chilled on ice before the next step. The 10 μL of fragmented mRNA, 0.5 μL of Random primer (Invitrogen), 0.5 μL of Oligo dT primer (Invitrogen), 0.5 μL of SUPERase-In (Ambion), 1 μL of dNTPs (10 mM), and 1 μL of DTT (10 mM) were heated at 50°C for 3 minutes. At the end of incubation, 5.8 μL of water, 1 μL of DTT (100 mM), 0.1 μL Actinomycin D (2 μg/μL), 0.2 μL of 1% Tween-20 (Sigma), and 0.2 μL of Superscript III (Invitrogen) were added and incubated in a PCR machine using the following conditions: 25°C for 10 minutes, 50°C for 50 minutes, and a 4°C hold. The product was then purified with Agentcourt RNAClean XP beads (Beckman Coulter) according to manufacturer's instruction and eluted with 10 μL nuclease-free water. The RNA/cDNA double-stranded hybrid was then added to 1.5 μL of Blue Buffer (Enzymatics), 1.1 μL of dUTP mix (10 mM dATP, dCTP, dGTP, and 20 mM dUTP), 0.2 μL of RNAse H (5 U/μL), 1.05 μL of water, 1 μL of DNA polymerase I (Enzymatics), and 0.15 μL of 1% Tween-20. The mixture was incubated at 16°C for 1 hour. The resulting dUTP-marked dsDNA was purified using 28 μL of Sera-Mag Speedbeads (Thermo Fisher Scientific), diluted with 20% PEG8000, 2.5 M NaCl to final of 13% PEG, eluted with 40 μL EB buffer (10 mM Tris-Cl (pH 8.5)), and frozen −80°C. The purified dsDNA (40 μL) underwent end repair by blunting, A-tailing and adapter ligation using barcoded adapters (NextFlex, Bioo Scientific). Libraries were PCR-amplified for 9 to 14 cycles, size selected by gel extraction, quantified by Qubit dsDNA HS Assay Kit (Thermo Fisher Scientific) and sequenced on a NextSeq 500.

### RNA sequencing data analysis

Data was analyzed by normalizing reads per million and using principal components analysis to determine genes with the highest PC1 (+) scores and lowest PC1 (-) scores between *dfm1*Δ +GAL$_{pr}$-Hmg2-GFP and every other strain tested. From this list, we used the top 100 genes with the highest (+) and lowest (-) PCA1 values, and cross referenced those to the normalized transcript per million reads value for each gene and removed genes that were not expressed at either a higher (for +PCA1 values) or lower (for -PCA1 values) reads per million level than all other conditions that were sequenced. Then, this list of up-regulated and down-regulated genes was used for GO analysis using http://geneontology.org/.

### Fluorescence microscopy

To prepare cells, overnight cultures were diluted to approximately 0.20 OD in minimal media lacking uracil (-URA). After growing approximately 3 hours, samples were pelleted and washed with dH2O before being resuspended in 80 uL of media to be used for imaging. Fluorescence microscopy was accomplished using a CSU-X1 Spinning Disk (Yokogawa) confocal microscope at the Nikon Imaging Center on the UCSD campus. Samples were analyzed to measure the fraction of GFP in puncta.

### Microscopy quantification and analysis

Microscopy images (16-bit) were analyzed using the Fiji distribution of ImageJ2 [81] and data was compiled in the RStudio integrated development environment of R [82]. Briefly, cell

outlines were segmented by uploading brightfield images of each field of view to the online YeastSpotter tool [83]; available at http://yeastspotter.csb.utoronto.ca/. Then, fluorescent micrographs from the 488 channel were maximum-projected for all z-slices. To identify Hmg2 puncta in the 488 channel, a 20-pixel median filter was subtracted from each max-Z projected image. Then, any fluorescent Hmg2 signal above a gray value of 750 was thresholded as a "puncta." This threshold value was applied to all fields of view, regardless of genotype, and was determined after manually comparing the puncta calls for representative images of each geno-type using different threshold values. The fraction of Hmg2 in these bright puncta relative to total Hmg2 in the cell was quantified by summing the integrated density of all puncta in a cell and dividing by the total integrated density of the cell. All ImageJ macros and Rscripts used in this analysis, as well as more detailed methods, are available in a GitHub repository from 3 March 2022 (https://github.com/LiviaSongster/yeast-fluor-percent-puncta). All statistical analy-sis was performed using GraphPad Prism version 8.0.

## Detergent solubility assay

ER microsomes were isolated by centrifuging and pelleting 15 OD of yeast in log phase growth. Pellets were resuspended in MF buffer (20 mM Tris (pH 7), 100 mM NaCl, 300 mM sorbitol) with protease inhibitors (PIs) (1 mM phenylmethylsulfonyl fluoride, 260 µM 4-(2-aminoethyl) benzenesulfonyl fluoride hydrochloride, 100 µM leupeptin hemisulfate, 76 µM pepstatin A, 5 mM aminocaproic acid, 5 mM benzamidine, and 142 µM N-tosyl-l-phenylalanine chloro-methyl ketone) and 0.5 mM lysis beads were added to each sample. Samples were vortexed 6 times in 1-minute intervals, with 1-minute on ice in between. Lysed cells were transferred to new microcentrifuge tube and samples were clarified by spinning at 1,500× for 5 minutes at 4°C. Microsomes were separated by centrifuging clarified lysate at 14,000 × $g$ for 1 minute. Fractions were incubated on ice in the presence or absence of 1% DDM for 1 hour. The mix-ture was then centrifuged at 14,000 × $g$ for 30 minutes at 4°C, and the detergent soluble frac-tion (i.e., the supernatant) was precipitated with 20% trichloroacetic acid (TCA) on ice for 30 minutes and then centrifuged at 14,000 × $g$ for 30 minutes to get a pellet of the soluble protein. Proteins from both the soluble and insoluble fractions were resuspended in sample buffer and resolved by SDS-PAGE.

## Co-Immunoprecipitation

Yeast were grown to mid log phase in minimal media, and 15 OD equivalents were pelleted, washed in water, and resuspended in 240 µl lysis buffer (0.24 M sorbitol, 1 mM EDTA, 20 mM KH2PO4/K2HPO4 (pH 7.5)) with PIs. Acid-washed glass beads were added up to the menis-cus. Cells were lysed on a multivortexer at 4°C for six to eight 1-minute intervals with 1 minute on ice in between each lysis step. The lysates were transferred to a new tube, and lysates cleared with 5-second pulses of centrifugation. Microsomes were pelleted from cleared lysates by cen-trifugation at 14,000 × $g$ for 5 minutes. Microsome pellets were washed once in XL buffer (1.2 M sorbitol, 5 mM EDTA, 0.1 M KH2PO4/K2HPO4 (pH 7.5)) and resuspended in XL buffer.

Samples were then solubilized by the addition of detergent solution at 10× the desired final concentration in XL buffer (final concentration of 1% DDM). Preparations with detergent were incubated at 4°C for 1 hour with rocking and then repeatedly pipetted up and down. Finally, samples were cleared by centrifugation in a benchtop microcentrifuge for 15 minutes at 16,000 $g$. The supernatants were then separated by ultracentrifugation at 89,000 RPM for 15 minutes, and the supernatant was incubated overnight with 10 µL of equilibrated GFP-Trap agarose (ChromoTek, Hauppauge, New York) at 4°C. The next day, the GFP-Trap agarose beads were combined to 1 tube, washed once with non-detergent IP buffer, washed once more

with IP wash buffer, and resuspended in 100 μL of 2xUSB. Samples were resolved on 8% SDS-PAGE and immunoblotted for Hmg2-GFP with anti-GFP, Dfm1-HA with anti-HA, and anti-Sec61 antibody.

## Cycloheximide-chase assay

Cycloheximide chase assays were performed as previously described [6]. Cells were grown to log-phase ($OD_{600}$ 0.2 to 0.3) and cycloheximide was added to a final concentration of 50 μg/mL. At each time point, a constant volume of culture was removed and lysed. Lysis was initiated with addition of 100 μL SUME with PIs and glass beads, followed by vortexing for 4 minutes. Approximately 100 μL of 2xUSB was added followed by incubation at 55°C for 10 minutes. Samples were clarified by centrifugation and analyzed by SDS-PAGE and immunoblotting.

## In vivo ubiquitination assay

Cells were grown to log phase ($OD_{600}$ 0.3 to 0.6) and 15 ODs of cells were pelleted. Cells were resuspended in $H_20$, centrifuged and lysed with the addition of 0.5 mM glass beads and 400 μL of XL buffer (1.2 M sorbitol, 5 mM EDTA, 0.1 M $KH_2PO_4$, final pH 7.5) with PIs followed by vortexing in 1-minute intervals for 6 to 8 minutes at 4°C. Lysates were combined and clarified by centrifugation at 2,500 g for 5 minutes. Approximately 100 μL clarified lysate was resuspended in 100 μL SUME (1% SDS, 8 M Urea, 10 mM MOPS (pH 6.8), 10 mM EDTA) with PIs and 5 mM N-ethyl maleimide (NEM, Sigma) followed by addition of 600 μL immunoprecipitation buffer (IPB) (15 mM Na2HPO4, 150 mM NaCl, 2% Triton X-100, 0.1% SDS, 0.5% deoxycholate, and 10 mM EDTA (pH 7.5)) with PIs and NEM. A total of 15 μL of rabbit polyclonal anti-GFP antisera (C. Zuker, University of California, San Diego) was added to the samples for immunoprecipitation (IP) of Hmg2-GFP. Samples were incubated on ice for 5 minutes, clarified at 14,000 *g* for 5 minutes, and removed to a new eppendorf tube and incubated overnight at 4°C. Approximately 100 μL of equilibrated Protein A-Sepharose in IPB (50% w/v) (Amersham Biosciences) was added and incubated for 2 hours at 4°C. Proteins A beads were washed twice with IPB and washed once more with IP wash buffer (50 mM NaCl, 10 mM Tris), aspirated to dryness, resuspended in 2× urea sample buffer (8 M urea, 4% SDS, 1 mM DTT, 125 mM Tris (pH 6.8)), and incubated at 55°C for 10 minutes. IPs were resolved by 8% SDS-PAGE, transferred to nitrocellulose, and immunoblotted with monoclonal anti-ubiquitin (Fred Hutchinson Cancer Center, Seattle) and anti-GFP (Clontech, Mountain View, California). Goat anti-mouse (Jackson ImmunoResearch, West Grove, Pennsylvania) and goat anti-rabbit (Bio-Rad) conjugated with horseradish peroxidase (HRP) recognized the primary antibodies. Western Lightning Plus (Perkin Elmer, Watham, Massachusetts) chemiluminescence reagents were used for immunodetection.

## Western blot quantification

Western blot images were quantified using ImageJ/Fiji. Band intensities were measured from high resolution TIF files of western blot images acquired from a BioRad Chemidoc Imager. Data analysis was done using Prism8 (GraphPad).

## Supporting information

**S1 Fig. Hmg2-GFP microscopy puncta are unaffected by Dfm1. (A and B)** WT and *cdc48-2* strains were grown into log-phase at 30°C and degradation was measured by cycloheximide chase (CHX). After CHX addition, cells were lysed at the indicated times, and analyzed by

SDS-PAGE and immunoblotted for Pdr5*-HA with α-HA and Ste6*-GFP with α-GFP. Three biological replicates ($N$ = 3). **(C)** Steady-state levels of Dfm1 and corresponding Dfm1 mutants from *dfm1Δ* cells containing GAL$_{pr}$-HMG2-GFP that were used for growth assays in Fig 2. Cells were analyzed by SDS-PAGE and immunoblotted with α-HA. Three biological replicates ($N$ = 3). **(D)** Representative confocal microscopy images of Hmg2-GFP in *dfm1Δ* cells with add-back of EV, WT DFM1, and DFM1-5Ashp. Five biological replicates were imaged, and 3 images were taken of each strain ($N$ = 5). **(E)** Fraction of Hmg2-GFP in puncta for *dfm1Δ* cells with add-back of WT DFM1, EV, and DFM1-5Ashp. Each dot represents an individual cell. **(F)** Number of puncta per cell for *dfm1Δ* cells with add-back of WT DFM1, EV, and DFM1-5Ashp. Each dot represents an individual cell. Data information: The data underlying this figure can be found in Table M and N in S1 Data (Sheet 4).
(TIF)

**S2 Fig. Transcriptional changes in membrane protein-stressed *dfm1Δ* cells. (A)** *pdr5Δ*, *pdr5Δdfm1Δ*, and *pdr5Δhrd1Δ* cells containing either GAL$_{pr}$-HMG2-GFP or EV were compared for growth by dilution assay. Each strain was spotted 5-fold dilutions on glucose or galactose-containing plates to drive HMG2-GFP overexpression, and plates were incubated at 30˚C. Three biological replicates and 2 technical replicates ($N$ = 3). **(B)** Principal component 1 (PC1) and principal component 2 (PC2) values of each of the 2 biological replicates ($N$ = 2) of RNA-seq samples for *pdr5Δ*, *dfm1Δpdr5Δ*, and *hrd1Δpdr5Δ* cells containing either GAL$_{pr}$-HMG2-GFP or EV. **(C)** PC1 and PC2 of sorted top 100 highest PC1 value genes from both replicates of *dfm1Δpdr5Δ* cells containing GAL$_{pr}$-HMG2-GFP. Red dots indicate Rpn4 target genes. Table indicates up-regulated genes that are targeted by Rpn4. **(D)** Top 10 GO terms and their enrichment factor for the set of 100 up-regulated genes with the highest PC1 scores. **(E)** Top 10 GO terms and their enrichment factor for the set of 100 down-regulated genes with the lowest PC1 scores. Data information: The data underlying this figure can be found in Table O–Q in S1 Data (Sheet 5) and Table R (Sheet 6).
(TIFF)

**S3 Fig. Misfolded membrane protein stress in *dfm1Δ* cells does not affect Hac1 splicing. (A)** PCR products of spliced and unspliced Hac1 transcripts. *pdr5Δ* cells containing GAL$_{pr}$-Hmg2-6MYC, GAL$_{pr}$-Ste6*-GFP, GAL$_{pr}$-CPY*-HA, or EV were treated with 0.2% galactose and 2 µg/mL tunicamycin (+) or an equivalent volume of DMSO (-). RNA was extracted from cells and cDNA was generated and used as a template for PCR. uHac1 represents unspliced Hac1 transcripts and sHac1 represents spliced Hac1. **(B)** Hac1 splicing assay as in (A) except using *dfm1Δ* cells. Data information: Images are representative of 3 biological replicates ($N$ = 3).
(TIF)

**S4 Fig. *Rpn4Δ* toxicity is specific to misfolded membrane proteins. (A)** WT, *dfm1Δ*, and *rpn4Δ* cells containing either GAL$_{pr}$-STE6*-GFP or EV were compared for growth by dilution assay. Each strain was spotted 5-fold dilutions on glucose or galactose-containing plates to drive Ste6*-GFP overexpression, and plates were incubated at 30˚C. **(B)** Dilution assay as depicted in (A) except using cells containing GAL$_{pr}$-CPY*-HA or EV. **(C)** Dilution assay as described in (A) except using WT, *dfm1Δ*, and *pdr1Δ* cells containing either GAL$_{pr}$-HMG2-GFP or EV. Data information: All dilution growth assays were performed in 3 biological and 2 technical replicates ($N$ = 3).
(TIF)

**S5 Fig. *Ubp6Δ* toxicity is specific to misfolded membrane proteins. (A)** Western blot of monomeric ubiquitin in WT, *dfm1Δ*, and *hrd1Δ* cells. Anti-ubiquitin was used to blot for

ubiquitin and anti-PGK1 was used to blot for PGK1 as a loading control. **(B)** Quantification of western blots from (A). Each strain was normalized to PGK1 and the monomeric ubiquitin quantification of WT was used to normalize all strains. **(C)** WT, *dfm1Δ*, and *ubp6Δ* cells containing either GAL<sub>pr</sub>-STE6*-GFP or EV were compared for growth by dilution assay. Each strain was spotted 5-fold dilutions on glucose or galactose-containing plates to drive Ste6*-GFP overexpression, and plates were incubated at 30˚C. **(D)** Dilution assay as depicted in (C) except using cells containing GAL<sub>pr</sub>-CPY*-HA or EV. Data information: All dilution growth assays were performed in 3 biological and 2 technical replicates (*N* = 3). For (B), all data are mean ± SEM, 3 biological replicates (*N* = 3); statistical significance is displayed as two-tailed unpaired *t* test, ns, not significant. The data underlying this figure can be found in Table S (Sheet 7).
(TIF)

**S6 Fig. Not all deubiquitinates mediate misfolded membrane toxicity and toxicity is specific to misfolded membrane proteins.** **(A)** WT, *dfm1Δ*, and *doa4Δ* cells containing either GAL<sub>pr</sub>-HMG2-GFP or EV were compared for growth by dilution assay. Each strain was spotted 5-fold dilutions on glucose or galactose-containing plates to drive Hmg2-GFP overexpression, and plates were incubated at 30˚C. **(B)** Dilution assay as in (A) except in *dfm1Δ*, *ubp2Δ*, *ubp5Δ*, *miy1Δ*, *ubp8Δ*, *miy2Δ*, *otu2Δ*, *ubp1Δ*, *ubp11Δ*, *ubp7Δ*, and *ubp3Δ* cells. **(C)** WT, *dfm1Δ*, *ubp6Δ*, *doa4Δ*, *ubp9Δ*, and *ubp14Δ* cells containing either GAL<sub>pr</sub>-ΔssCPY*-Myc or EV were compared for growth by dilution assay. Each strain was spotted 5-fold dilutions on glucose or galactose-containing plates to drive Hmg2-GFP overexpression, and plates were incubated at 30˚C. Data information: All dilution growth assays were performed in 3 biological and 2 technical replicates (*N* = 3).
(TIF)

**S7 Fig. Genetic interactions between Dfm1, Rpn4, and Ubp6 in resolving misfolded membrane protein toxicity.** **(A)** *dfm1Δ*, *dfm1Δrpn4Δ*, *dfm1Δubp6Δ*, and *rpn4Δubp6Δ* cells containing either GAL<sub>pr</sub>-HMG2-GFP or EV were compared for growth by dilution assay. Each strain was spotted 5-fold dilutions on glucose or galactose-containing plates to drive Hmg2-GFP overexpression, and plates were incubated at 30˚C. **(B)** Dilution assays as depicted in **(A)** except using cells containing GAL<sub>pr</sub>-STE6*-GFP. **(C)** Dilution assays as depicted in (A) except using cells containing GAL<sub>pr</sub>-CPY*. **(D)** *dfm1Δ*, *rpn4Δ*, and *ubp6Δ* cells containing either GAL<sub>pr</sub>-Hmg2-GFP or EV and GAL<sub>pr</sub>-Dfm1-10xHis or EV were compared for growth by dilution assay. Each strain was spotted 5-fold dilutions on glucose or galactose-containing plates to drive Hmg2-GFP and Dfm1-10xHis overexpression, and plates were incubated at 30˚C. **(E)** Dilution assay as described in **(A)** except using *rpn4Δ* and *ubp6Δ* cells containing either GAL<sub>pr</sub>-Hmg2-GFP, GAL<sub>pr</sub>-Hmg2 (K6R)-GFP, GAL<sub>pr</sub>-Hmg2 (K357R)-GFP, GAL<sub>pr</sub>-Hmg2 (K6R and K357R)-GFP, or EV. Data information: All dilution growth assays were performed in 3 biological and 2 technical replicates (*N* = 3).
(TIF)

**S1 Table. Plasmids used in this study.**
(DOCX)

**S2 Table. Yeast strains used in this study.**
(DOCX)

**S1 Raw images. Blot-gel data file.**
(PDF)

**S1 Data. Raw data files for Figs 5–7 and S1 and S2 and S5.**
(XLSX)

## Acknowledgments

We thank Tom Rapoport (Harvard Medical School), Davis Ng (National University of Singapore), Randy Schekman (University of California, Berkeley), Susan Michaelis (John Hopkins University), and Jeff Brodsky (University of Pittsburgh) for providing plasmids and antibodies. We also thank the Neal lab members for their positive reinforcement, in depth discussions, and technical assistance.

## Author Contributions

**Conceptualization:** Rachel Kandel, Jasmine Jung, Della Syau, Sascha Duttke, Christopher Benner, Sonya E. Neal.

**Data curation:** Rachel Kandel, Jasmine Jung, Della Syau, Tiffany Kuo, Livia Songster, Casey Horn, Claire Chapman, Analine Aguayo, Sascha Duttke, Sonya E. Neal.

**Formal analysis:** Rachel Kandel, Jasmine Jung, Tiffany Kuo, Livia Songster, Casey Horn, Claire Chapman, Analine Aguayo, Sascha Duttke, Christopher Benner, Sonya E. Neal.

**Investigation:** Rachel Kandel, Della Syau, Christopher Benner, Sonya E. Neal.

**Methodology:** Rachel Kandel, Sascha Duttke, Sonya E. Neal.

**Resources:** Sonya E. Neal.

**Supervision:** Rachel Kandel, Sonya E. Neal.

**Validation:** Rachel Kandel, Della Syau, Tiffany Kuo, Livia Songster, Casey Horn, Claire Chapman, Sascha Duttke, Christopher Benner, Sonya E. Neal.

**Visualization:** Jasmine Jung, Sonya E. Neal.

**Writing – original draft:** Rachel Kandel, Sascha Duttke, Christopher Benner, Sonya E. Neal.

**Writing – review & editing:** Rachel Kandel, Sonya E. Neal.

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
