## [Editor Report · Decision Letter 0]

11 Aug 2022

Dear Dr neal, 

Thank you for submitting your manuscript entitled "Derlin Dfm1 employs a chaperone function to resolve misfolded membrane protein stress" for consideration as a Research Article by PLOS Biology. I am handling your manuscript for my colleague Ines whilst she is out of the office. Please accept my apologies for the delay in getting back to you whilst we consulted with an academic editor about your submission. 

Your manuscript has now been evaluated by the PLOS Biology editorial staff, as well as by an academic editor with relevant expertise, and I am writing to let you know that we would like to send your submission out for external peer review.

During our evaluation, we noted that Figure 1A in the current manuscript appears to be the same image from your previous publication in iScience (Neal et al, 2020, PMID 32891886). Due to copyright, we ask that you please either (a) remove the figure from the manuscript and cite the previous work in the manuscript text or (b) provide a different image of the same substrate toxicity experiment. 

Before we can send your manuscript to reviewers, we need you to complete your submission by providing the metadata that is required for full assessment. To this end, please login to Editorial Manager where you will find the paper in the 'Submissions Needing Revisions' folder on your homepage. Please click 'Revise Submission' from the Action Links and complete all additional questions in the submission questionnaire.

Once your full submission is complete, your paper will undergo a series of checks in preparation for peer review. After your manuscript has passed the checks it will be sent out for review. To provide the metadata for your submission, please Login to Editorial Manager (https://www.editorialmanager.com/pbiology) within two working days, i.e. by Aug 13 2022 11:59PM.

Kind regards,

Richard

Richard Hodge, PhD

Associate Editor, PLOS Biology

rhodge@plos.org

On behalf of:

Ines Alvarez-Garcia, PhD

---

## [Decision Letter · Decision Letter 1]

29 Sep 2022

Dear Dr Neal,

Thank you for your patience while your manuscript entitled "Derlin Dfm1 employs a chaperone function to resolve misfolded membrane protein stress" was peer-reviewed at PLOS Biology. It has now been evaluated by the PLOS Biology editors, an Academic Editor with relevant expertise, and by two independent reviewers. 

The reviews are attached below. As you will see, the reviewers find the conclusions novel and significant for the field, but they also raise several issues that should be addressed to strengthen some of the findings. Reviewer 1 would like you to clarify several points and add missing references, whereas Reviewer 2 suggests additional experiments to confirm the findings. After discussing the reviews with the Academic Editor, we would like to invite you to submit a revision that addresses all the issues raised by the reviewers

Given the extent of revision needed, we cannot make a decision about publication until we have seen the revised manuscript and your response to the reviewers' comments. Your revised manuscript is likely to be sent for further evaluation by all or a subset of the reviewers.

**IMPORTANT - SUBMITTING YOUR REVISION**

3. Resubmission Checklist

a) *PLOS Data Policy*

b) *Published Peer Review*

d) *Blurb*

Please also provide a blurb which (if accepted) will be included in our weekly and monthly Electronic Table of Contents, sent out to readers of PLOS Biology, and may be used to promote your article in social media. The blurb should be about 30-40 words long and is subject to editorial changes. It should, without exaggeration, entice people to read your manuscript. It should not be redundant with the title and should not contain acronyms or abbreviations. For examples, view our author guidelines: https://journals.plos.org/plosbiology/s/revising-your-manuscript#loc-blurb

Sincerely,

Ines

--

Ines Alvarez-Garcia, PhD

Senior Editor

PLOS Biology

Reviewers' comments

Rev. 1:

In this manuscript submitted to PLOS Biology (PBIOLOGY-D-22-01696R1), Kandel et al. examine the contribution of the S. cerevisiae Rhomboid pseudoprotease, Dfm1 in the solubilization of misfolded integral membrane proteins. The role of Dfm1 in preparing integral membrane substrates for retrotranslocation via substrate engagement, membrane thinning, and Cdc48 recruitment has been established by a series of well-executed studies. Using a substrate toxicity assay and a plethora of constructs/tools to study Dfm1 function, growth assays, and Western Blotting, this report aims to establish a hitherto unappreciated Dfm1 chaperone-like activity. Moreover, while it has long been known that the expression of misfolded integral membrane proteins can cause toxicity in certain yeast genetic backgrounds (knockout strains), the mechanism has remained elusive. Therefore, in this manuscript, using genetic tools, the authors try to dissect the cellular event that is eliciting the toxicity which is specific to integral membrane ERAD substrates but not the lumenal substrate (CPY*). Overall, the data are presented in a clear and logical fashion and the quality of the data is sufficient to make the arguments convincing. The paper presents new and surprising data on all fronts and will be an excellent addition to the field. It was very pleasing to read, as it methodically answered each new question opened by the previous paragraph. The study also raised new and interesting questions to explore. There are just a few minor concerns with figures, references, and framing that should be fixed prior to acceptance for publication.

Text Concerns

1. Line 52, more primary references should be added, Bays Nat Cell Bio 2001: Meyer EMBO J 2000: Rabinovich Mol Cell Biol 2002: Ye and Rapoport Nature 2001

2. Line 25 vs Line 490 clear up the number of integral membrane proteins (should be closer to ¼ as initially stated)

3. There should be a reference included for the end of line 73.

4. There should be a reference included for the end of 87.

5. Line 134, sentence ending ER E3 ligases. A reference should be added or data should be added to the supplement for proof of the statment. Often ERAD substrates can require the action of two or more E3 ligases, so to say that the substrates are not ubiquitinated in single deletions is likely misleading. Perhaps it could be reworded to say reduced ubiquitination…

6. Line 299, I do not recall seeing any discussion on what the source of the variable transcriptional changes could be.

7. Line 328, reference should be added for Burns FEBS letter 2021, for role of Rpn4.

Figure/Experimental concerns

1. Line 137-143. The use of the cdc48-2 allele is clever, but if cdc48 was truly inactivated, then the strain should not divide making the spot test assay impossible. I saw that the permissive temperature of 30 degrees was used, but this is in opposition to most of the papers which use this cdc48 alleles at 37 degrees or higher (Simoes eLife 2018;Gallagher J Cel Sci 2014:Hsieh PLOS One 2011, to name a few) . While it has been shown for Hmg2 that 30 degrees is sufficient to induce an ERAD defect by flow cytometry (Hampton lab) this has not been shown for Pdr5* and Ste6p*. Some language should be added to hedge bets about the impact of the 30 degrees on the substrates. Better yet, if you had some experimental data to show that 30 degrees has an impact for the degradation or retrotranslocation of these substrates, that would be great to include.

2. It is not unexpected that CFTR and F508del have similar phenotypes as this as been observed by others but not formally published, but in a dissertation somewhere.

3. In Figure 6G and the text starting line 196. You discuss all 3 derlins, but then only show data that two rescued. Was derlin 3 not tested, or did it not rescue? If it was not tested, please state.

4. For the detergent solubility assay, it is difficult to know, the data are so clean it is hard to believe! All or nothing. However, a thought occurred as to once the substrate is released from the cell, is Dfm1 still associated with the substrate in DDM, e.g. would they co-IP in this detergent? If that data is available, please include or discuss findings as data not shown. If so, that would explain why Dfm1 can still impart its effect post-lysis. If Dfm1 falls off during the lysis procedure, one must wonder, how did Dfm1 permanently alter the solubility of the substrate, such at it stays in the supe post-lysis?

5. ¬The flow cytometry data for UPR activation presented in figure S3 is a little confusing to me. In the text you try to make the point that CPY* exacerbates UPR activation in Dfm1 delta cells, but that Hmg2 does not in Der1delta cells. Then I look at the AU between the left column and right column in Fig. S3 and the absolute values can be quite different between the pdr5 strain and the dfm1 strain. It looks like dfm1 knockout is having some impact on baseline ability to activate the UPR because For Ste6p* for example, the pdr5 AU in gal+TM is 3K, and in the dfm1 the gal + TM condition is 30K. Perhaps fold change induction would a better way to display the data. Or some more text to describe why these differences are present and what they mean.

6. Figure 7A, the data for the dfm1delta over expressing Hmg2 and Ste6p* shows growth when in every other figure it shows no growth. This makes the data for this figure difficult to interpret. The figure should be replaced with data that more accurately matches the phenotype present in other growth assays.

Rev. 2:

This is an interesting manuscript, which describes a comprehensive study that aims to define a new role of Dfm1 in solubilizing aggregates formed by membrane proteins in ER.

This is a follow-up study to one that was published by the same group last year in Cell Reports that identified specific residues in Dfm1 that crucial for the recognition of ubiquitinated ERAD membrane substrates and their retrotranslocation to ER.

In this manuscript, the authors investigate the additional function of Dfm1, which leads to the solubilization of ubiquitinated membrane protein aggregates. This function is Cdc48 independent.

The paper is very well written, however, some of the parts are confusing and the rationale of the experiments as well as their outcomes are not very clear in the context of the whole story. It is especially true for the transcriptomic analysis that was done using different mutant strains than in the rest of the manuscript. In addition, some figures are not cited correctly or missing.

Moreover, the main conclusions of this study are based on growth defect phenotypes shown using dilution spot assays. This is a valid approach in the field, however, since ~80-90% of the main figures are dilution spot assays without any replicates, additional experiments should be provided, such as growth in medium or others.

The main conclusion is that Dfm1 prevents the formation of aggregates formed by membrane proteins, substrates of ERAD, is solid and well proven by the phenotypic and biochemical study, including solubility test and UPR activation analysis. However, the second statement that dfm1 deletion "resulting in compromised proteasome function and a depletion of monomeric ubiquitin" is based mainly on phenotypic studies and Figs 7D-E only, which shows not a very significant difference in mono-Ubiquitin.

The fact that many other single KO strains led to the similar phenotype as dfm1 delta strain suggests that this phenomenon might be broader and can be rooted in dfm1-indirect or parallel pathways (as also suggested by the authors at some parts).

Moreover, in all bar figures, it is not clear how the significance of the t-test was defined (what p-value cutoff was used, what are the parameters of t-test were used). Seems that some of the changes are not statistically significant (6B 5h induction, 7E).

In addition, Fig 2A was already published in Cell Reports and was also used here. This figure should be modified and additional labeling of the DFM1 motifs should be added.

To be useful for the community, the authors should address the following concerns.

Major comments:

1. Chaperone-activity of dfm1 (line 228). The paper does not show any direct chaperone activity of DFM1, thus, this phrase and the related statements are confusing. The aggregation clearance effect might be direct or indirect. This should be discussed and the term "chaperone" should not be used in this context.

Characterization of the anti-aggregation activity of DFM1:

Fig 3 A-B and 4B are one of the most important figures in this manuscript, however, no statistics was shown.

Moreover, any idea why there are large differences between "total" and "pellet" or "sup" fractions, especially in Fig 4 D? Alternatively, Fig. 4C and E suggest that there is the degradation of the tested proteins (high level in T and a small level of proteins in S). Also, differences in levels of Hmg2 in 4A might point to this.

The authors tried to detect aggregates in cells using confocal microscopy, however, this analysis was not very successful as also mentioned by the authors (S1). The provided images in S1 are very strange. It is unclear where are the bounders of the cells, seems like these are dead cells or may be vacuoles. GFP-Hmg2 was successfully used by others, for example in Schafer et al., EMBO 2019, Hampton et al., PNAS 1996. An improvement of these assays in this study can be very useful.

2. The authors showed very nicely that different substrates, Hmg2, Pdr5* , Ste6*, CFTR variants undergo aggregation in the dfm1 delta strains, and that it does not depend on the Cdc48 binding domain. The authors also tested levels of free ubiquitin in the cells expressing Hmg2 and detected a slight decrease in the dfm1 cells relative to hrd1 delta strains (not clear if it is really statistically significant due to large errors). Together with phenotypic studies that show growth defects in hrd and doa delta strains, the authors propose that "membrane substrates are ubiquitinated in delta dfm1 strains but not in delta hrd1, doa1" (line 132) - there is no experimental support for this in this paper.

Later on, the authors show that the same growth defect is common to dfm1 delta, rpn4 delta and pdr1 delta strains. These can be due to related or non-related pathways.

3. Lines 178 - 184 . Are the expression levels of the Dfm1 variants similar to the wt?

4. lines 196-204 - Fig 6G is incorrect. I cannot find the related figure to this statement. Moreover, the relevance of this part is not very clear or developed.

5. lines 263-270- Fig 2A-H does not relate to this sentence. No time dependence was shown in any figure.

The statement "Unexpectedly, the level of UPR activation was much higher in dfm1Δ cells expressing CPY* than in pdr5Δ cells" is not supported by the S2E-F figures. It looks the opposite. The background in delta dfm1-CPY* is high with or without induction of CPY*. Maybe it is true for shorter incubation time? But it is not shown in the provided figures.

Line 276 " increases sensitivity to other stresses" - it is not tested here, so it is very speculative at this point.

6. Transcriptomic analysis of dfm1 delta cells expressing Hmg2. I am not sure this part adds to the story, I found it very confusing and not developed. I think it can be removed and used in another manuscript.

There is inconsistency in the used strains: in the results and methods session- dfm1 delta+ Hmg2 cells but in the figure - double mutant, dfm1 and pdr5 delta. If indeed the analysis was done in the double mutant strain then it is not relevant to results discussed previously and additional controls should be added (growth phenotype, solubility test and others).

Moreover, an additional table should be used to support the overlap between Rpn4 targets and the identified dfm1-related genes (lines 318- 319)

7. Fig 7A : the phenotype in dfm1 delta strain is less strong than in Fig 1. Thus, to compare different strains, consider conducting a complementary test, such as growth in medium or others.

In addition, Fig 7A cannot be a basis for the statement in line 387 : "This indicates that growth stress in dfm1Δ cells is dependent upon ubiquitination of the accumulated misfolded membrane protein." No ubiquitination analysis was provided here.

Fig 7B - it is still not clear to me why the expression of the Hmg2 double mutant K6R-K357R has such a strong recovery effect (similar to wt growth) while single mutants lead to significant cell death. Moreover, the quality of figure S8 is not very good. Is it possible to use CFU values?

Minor comments:

1. Fig 2A. low resolution, very difficult to distinguish between different motifs colored by very close colors. The figure legend does not correspond to the colors used in the figure. Should be modified and the motifs should be labeled better. Moreover, this figure is already used in another publication.

In addition, motif GxxxG (line 175) is not labeled in the figure. Shp1 domain should be shown in 2A.

2. Fig 2: there is no consistency between the B, C and E panels in the terms of order (+DFM1 followed by -DFM1).

3. line 260 - what does "optical reporter" mean? Consider rephrasing

4. Fig 5. Panel B is after D

5. Fig 5 - what are the p-values (and what test) were used for the functional enrichment?

6. line 334 . Pre6-GFP - should be explained what Pre6 is and why it was used here.

7. Fig 6B and C - details about the t-test analysis are needed. Doesn't look very significant (pink and green bars). 6D-F : dfm1 delta should be shown as well.

The colors are very confusing because of the previous panels. Consider do not use colors in the SUS-GFP bars as this is a different experiment and can be confused with the background of cells (expressing Hmg2, CPY* etc).

---

## [Decision Letter · Decision Letter 2]

7 Nov 2022

Dear Dr Neal,

Thank you for your patience while we considered your revised manuscript entitled "Derlin Dfm1 employs a chaperone-like function to resolve misfolded membrane protein stress" for publication as a Research Article at PLOS Biology. This revised version of your manuscript has been evaluated by the PLOS Biology editors, the Academic Editor and one of the original reviewers. We were unable to obtain comments from the other reviewer, but the Academic Editor has checked the responses to this reviewer and was satisfied.

Based on the review (attached below), we are likely to accept this manuscript for publication, provided you satisfactorily address the data and other policy-related requests stated below.

In addition, we would like you to consider a suggestion to improve the title:

"The yeast derlin Dfm1 uses a chaperone-like function to resolve misfolded membrane protein stress"

We expect to receive your revised manuscript within two weeks. 

*Published Peer Review History*

*Press*

Sincerely,

Ines

--

Ines Alvarez-Garcia, PhD

Senior Editor

PLOS Biology

Fig. 5A-H; Fig. 6B-F; Fig. 7F; Fig. S1E-F; Fig. S2B-D and Fig. S5B

In addition, please amend the panel letters in Fig. S1 – there are two E panels.

We require the original, uncropped and minimally adjusted images supporting all blot and gel results reported in an article's figures or Supporting Information files. We will require these files before a manuscript can be accepted so please prepare and upload them now. Please carefully read our guidelines for how to prepare and upload this data: https://journals.plos.org/plosbiology/s/figures#loc-blot-and-gel-reporting-requirements

Reviewers' comments

Rev. 1:

I would like to thank the authors for their careful attention to my comments and suggestions, including edits to the text and inclusion of additional experimental data. I am quite please with the finished manuscript and now find it worthy of immediate publication with no further scientific revision. I am confident this manuscript makes a meaningful contribution to the field and will be broadly accessible to PLOS Biology readers.

---

## [Editor Report · Decision Letter 3]

7 Dec 2022

Dear Dr Neal,

Thank you for the submission of your revised Research Article entitled "Yeast Derlin Dfm1 employs a chaperone-like function to resolve misfolded membrane protein stress" for publication in PLOS Biology. On behalf of my colleagues and the Academic Editor, Ursula Jakob, I am delighted to say that we can in principle accept your manuscript for publication, provided you address any remaining formatting and reporting issues. These will be detailed in an email you should receive within 2-3 business days from our colleagues in the journal operations team; no action is required from you until then. Please note that we will not be able to formally accept your manuscript and schedule it for publication until you have completed any requested changes.

PRESS

Sincerely, 

Ines

--

Ines Alvarez-Garcia, PhD

Senior Editor

PLOS Biology
